# Depth Uncertainty in Neural Networks

**Javier Antorán**[*]
University of Cambridge
`ja666@cam.ac.uk`

**James Urquhart Allingham**[*]
University of Cambridge
`jua23@cam.ac.uk`

**José Miguel Hernández-Lobato**
University of Cambridge
Microsoft Research
The Alan Turing Institute
`jmh233@cam.ac.uk`

## Abstract

Existing methods for estimating uncertainty in deep learning tend to require multiple forward passes, making them unsuitable for applications where computational resources are limited. To solve this, we perform probabilistic reasoning over the depth of neural networks. Different depths correspond to subnetworks which share weights and whose predictions are combined via marginalisation, yielding model uncertainty. By exploiting the sequential structure of feed-forward networks, we are able to both evaluate our training objective and make predictions *with a single forward pass*. We validate our approach on real-world regression and image classification tasks. Our approach provides uncertainty calibration, robustness to dataset shift, and accuracies competitive with more computationally expensive baselines.

## 1 Introduction

Despite the widespread adoption of deep learning, building models that provide robust uncertainty estimates remains a challenge. This is especially important for real-world applications, where we cannot expect the distribution of observations to be the same as that of the training data. Deep models tend to be pathologically overconfident, even when their predictions are incorrect (Nguyen et al., 2015; Amodei et al., 2016). If AI systems would reliably identify cases in which they expect to underperform, and request human intervention, they could more safely be deployed in medical scenarios (Filos et al., 2019) or self-driving vehicles (Fridman et al., 2019), for example.

In response, a rapidly growing subfield has emerged seeking to build uncertainty aware neural networks (Hernández-Lobato and Adams, 2015; Gal and Ghahramani, 2016; Lakshminarayanan et al., 2017). Regrettably, these methods rarely make the leap from research to production due to a series of shortcomings. *1) Implementation Complexity:* they can be technically complicated and sensitive to hyperparameter choice. *2) Computational cost:* they can take orders of magnitude longer to converge than regular networks or require training multiple networks. At test time, averaging the predictions from multiple models is often required. *3) Weak performance:* they rely on crude approximations to achieve scalability, often resulting in limited or unreliable uncertainty estimates (Foong et al., 2019a).

In this work, we introduce Depth Uncertainty Networks (DUNs), a probabilistic model that treats the depth of a Neural Network (NN) as a random variable over which to perform

---

[*]equal contribution

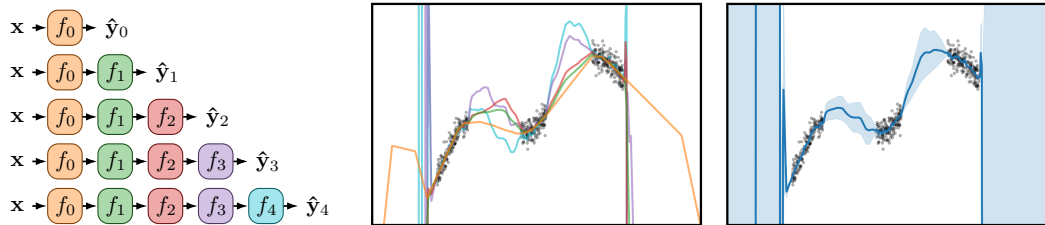

Figure 1: A DUN is composed of subnetworks of increasing depth (*left*, colors denote layers with shared parameters). These correspond to increasingly complex functions (*centre*, colors denote depth at which predictions are made). Marginalising over depth yields model uncertainty through disagreement of these functions (*right*, error bars denote 1 std. dev.).

inference. In contrast to more typical weight-space approaches for Bayesian inference in NNs, ours reflects a lack of knowledge about how deep our network should be. We treat network weights as learnable hyperparameters. In DUNs, marginalising over depth is equivalent to performing Bayesian Model Averaging (BMA) over an ensemble of progressively deeper NNs. As shown in Figure 1, DUNs exploit the overparametrisation of a single deep network to generate diverse explanations of the data. The key advantages of DUNs are:

1. *Implementation simplicity*: requiring only minor additions to vanilla deep learning code, and no changes to the hyperparameters or training regime.

2. *Cheap deployment*: computing exact predictive posteriors with a single forward pass.

3. *Calibrated uncertainty*: our experiments show that DUNs are competitive with strong baselines in terms of predictive performance, Out-of-distribution (OOD) detection and robustness to corruptions.

## 2   Related Work

Traditionally, Bayesians tackle overconfidence in deep networks by treating their weights as random variables. Through marginalisation, uncertainty in weight-space is translated to predictions. Alas, the weight posterior in Bayesian Neural Networks (BNNs) is intractable. Hamiltonian Monte Carlo (Neal, 1995) remains the gold standard for inference in BNNs but is limited in scalability. The Laplace approximation (MacKay, 1992; Ritter et al., 2018), Variational Inference (VI) (Hinton and van Camp, 1993; Graves, 2011; Blundell et al., 2015) and expectation propagation (Hernández-Lobato and Adams, 2015) have all been proposed as alternatives. More recent methods are scalable to large models (Khan et al., 2018; Osawa et al., 2019; Dusenberry et al., 2020). Gal and Ghahramani (2016) re-interpret dropout as VI, dubbing it MC Dropout. Other stochastic regularisation techniques can also be viewed in this light (Kingma et al., 2015; Gal, 2016; Teye et al., 2018). These can be seamlessly applied to vanilla networks. Regrettably, most of the above approaches rely on factorised, often Gaussian, approximations resulting in pathological overconfidence (Foong et al., 2019a).

It is not clear how to place reasonable priors over network weights (Wenzel et al., 2020a). DUNs avoid this issue by targeting depth. BNN inference can also be performed directly in function space (Hafner et al., 2018; Sun et al., 2019; Ma et al., 2019; Wang et al., 2019). However, this requires crude approximations to the KL divergence between stochastic processes. The equivalence between infinitely wide NNs and Gaussian processes (GPs) (Neal, 1995; de G. Matthews et al., 2018; Garriga-Alonso et al., 2019) can be used to perform exact inference in BNNs. Unfortunately, exact GP inference scales poorly in dataset size.

Deep ensembles is a non-Bayesian method for uncertainty estimation in NNs that trains multiple independent networks and aggregates their predictions (Lakshminarayanan et al., 2017). Ensembling provides very strong results but is limited by its computational cost. Huang et al. (2017), Garipov et al. (2018), and Maddox et al. (2019) reduce the cost of training an ensemble by leveraging different weight configurations found in a single SGD trajectory. However, this comes at the cost of reduced predictive performance (Ashukha et al., 2020). Similarly to deep ensembles, DUNs combine the predictions from a set of deep models. However, this set stems from treating depth as a random variable. Unlike ensembles,

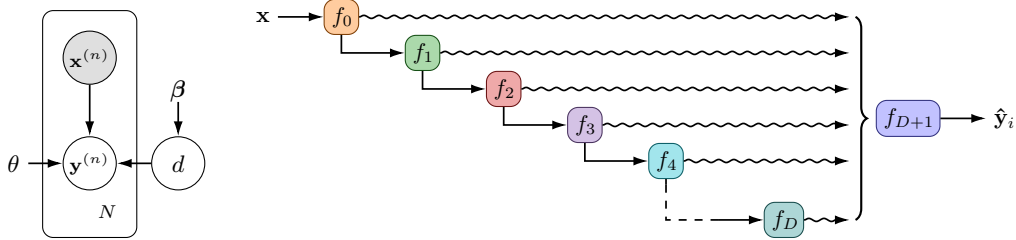

Figure 2: Left: graphical model under consideration. Right: computational model. Each layer's activations are passed through the output block, producing per-depth predictions.

BMA assumes the existence of a single correct model (Minka, 2000). In DUNs, uncertainty arises due to a lack of knowledge about how deep the correct model is. It is worth noting that deep ensembles can also be interpreted as approximate BMA (Wilson, 2020).

All of the above methods, except DUNs, require multiple forward passes to produce uncertainty estimates. This is problematic in low-latency settings or those in which computational resources are limited. Note that certain methods, such as MC Dropout, can be parallelised via batching. This allows for some computation time / memory usage trade-off. Alternatively, Postels et al. (2019) use error propagation to approximate the dropout predictive posterior with a single forward pass. Although efficient, this approach shares pathologies with MC Dropout. van Amersfoort et al. (2020) combine deep RBF networks with a Jacobian regularisation term to deterministically detect OOD points. Nalisnick et al. (2019c) and Meinke and Hein (2020) use generative models to detect OOD data without multiple predictor evaluations. Unfortunately, deep generative models can be unreliable for OOD detection (Nalisnick et al., 2019b) and simpler alternatives might struggle to scale.

There is a rich literature on probabilistic inference for NN structure selection, starting with the Automatic Relevance Detection prior (MacKay et al., 1994). Since then, a number of approaches have been introduced (Lawrence, 2001; Ghosh et al., 2019). Perhaps the closest to our work is that of Nalisnick et al. (2019a), which interprets dropout as a *structured shrinkage prior* that reduces the influence of residual blocks in ResNets. Conversely, a DUN can be constructed for any feed-forward neural network and marginalizes predictions at different depths. Similar work from Dikov and Bayer (2019) uses VI to learn both the width and depth of a NN by leveraging continuous relaxations of discrete probability distributions. For depth, they use a Bernoulli distribution to model the probability that any layer is used for a prediction. In contrast to their approach, DUNs use a Categorical distribution to model depth, do not require sampling for evaluation of the training objective or making predictions, and can be applied to a wider range of NN architectures, such as CNNs. Huang et al. (2016) stochastically drop layers as a ResNet training regularisation approach. On the other hand, DUNs perform exact marginalisation over architectures at train and test time, translating depth uncertainty into uncertainty over a broad range of functional complexities.

DUNs rely on two insights that have recently been demonstrated elsewhere in the literature. The first is that a single over-parameterised NN is capable of learning multiple, diverse, representations of a dataset. This is also a key insight for a subsequent work MIMO (Havasi et al., 2020). The second is that ensembling NNs with varying hyperparameters, in our case depth, leads to improved prediction robustness. Concurrently to our work, hyper-deep ensembles (Wenzel et al., 2020b) demonstrate this for a large range of hyperparameters.

## 3  Depth Uncertainty Networks

Consider a dataset $\mathfrak{D} = \{\mathbf{x}^{(n)}, \mathbf{y}^{(n)}\}_{n=1}^N$ and a neural network composed of an input block $f_0(\cdot)$, $D$ intermediate blocks $\{f_i(\cdot)\}_{i=1}^D$, and an output block $f_{D+1}(\cdot)$. Each block is a group of one or more stacked linear and non-linear operations. The activations at depth $i \in [0, D]$, $\mathbf{a}_i$, are obtained recursively as $\mathbf{a}_i = f_i(\mathbf{a}_{i-1})$, $\mathbf{a}_0 = f_0(\mathbf{x})$.

A forward pass through the network is an iterative process, where each successive block $f_i(\cdot)$ refines the previous block's activation. Predictions can be made at each step of this procedure by applying the output block to each intermediate block's activations: $\hat{\mathbf{y}}_i = f_{D+1}(\mathbf{a}_i)$. This

computational model is displayed in Figure 2. It can be implemented by changing 8 lines in a vanilla PyTorch NN, as shown in Appendix H. Recall, from Figure 1, that we can leverage the disagreement among intermediate blocks' predictions to quantify model uncertainty.

## 3.1 Probabilistic Model: Depth as a Random Variable

We place a categorical prior over network depth $p_{\boldsymbol{\beta}}(d) = \mathrm{Cat}(d|\{\beta_i\}_{i=0}^D)$. Referring to network weights as $\boldsymbol{\theta}$, we parametrise the likelihood for each depth using the corresponding subnetwork's output: $p(\mathbf{y}|\mathbf{x}, d{=}i; \boldsymbol{\theta}) = p(\mathbf{y}|f_{D+1}(\mathbf{a}_i; \boldsymbol{\theta}))$. A graphical model is shown in Figure 2. For a given weight configuration, the likelihood for every depth, and thus our model's Marginal Log Likelihood (MLL):

$$\log p(\mathfrak{D}; \boldsymbol{\theta}) = \log \sum_{i=0}^{D} \left( p_{\boldsymbol{\beta}}(d{=}i) \cdot \prod_{n=1}^{N} p(\mathbf{y}^{(n)}|\mathbf{x}^{(n)}, d{=}i; \boldsymbol{\theta}) \right), \tag{1}$$

can be obtained with a *single forward pass* over the training set by exploiting the sequential nature of feed-forward NNs. The posterior over depth, $p(d|\mathfrak{D}; \boldsymbol{\theta}) = p(\mathfrak{D}|d; \boldsymbol{\theta})p_{\boldsymbol{\beta}}(d)/p(\mathfrak{D}; \boldsymbol{\theta})$ is a categorical distribution that tells us about how well each subnetwork explains the data.

A key advantage of deep neural networks lies in their capacity for automatic feature extraction and representation learning. For instance, Zeiler and Fergus (2014) demonstrate that CNNs detect successively more abstract features in deeper layers. Similarly, Frosst et al. (2019) find that maximising the entanglement of different class representations in intermediate layers yields better generalisation. Given these results, using all of our network's intermediate blocks for prediction might be suboptimal. Instead, we infer whether each block should be used to learn representations or perform predictions, which we can leverage for ensembling, by treating network depth as a random variable. As shown in Figure 3, subnetworks too shallow to explain the data are assigned low posterior probability; they perform feature extraction.

## 3.2 Inference in DUNs

We consider learning network weights by directly maximising (1) with respect to $\boldsymbol{\theta}$, using backpropagation and the *log-sum-exp* trick. In Appendix B, we show that the gradients of (1) reaching each subnetwork are weighted by the corresponding depth's posterior mass. This leads to local optima where all but one subnetworks' gradients vanish. The posterior collapses to a delta function over an arbitrary depth, leaving us with a deterministic NN. When working with large datasets, one might indeed expect the true posterior over depth to be a delta. However, because modern NNs are underspecified even for large datasets, multiple depths should be able to explain the data simultaneously (shown in Figure 3 and Appendix B).

We can avoid the above pathology by decoupling the optimisation of network weights $\boldsymbol{\theta}$ from the posterior distribution. In latent variable models, the Expectation Maximisation (EM) algorithm (Bishop, 2007) allows us to optimise the MLL by iteratively computing $p(d|\mathfrak{D}; \boldsymbol{\theta})$ and then updating $\boldsymbol{\theta}$. We propose to use stochastic gradient variational inference as an alternative more amenable to NN optimisation. We introduce a surrogate categorical distribution over depth $q_{\boldsymbol{\alpha}}(d) = \mathrm{Cat}(d|\{\alpha_i\}_{i=0}^D)$. In Appendix A, we derive the following lower bound on (1):

$$\log p(\mathfrak{D}; \boldsymbol{\theta}) \geq \mathcal{L}(\boldsymbol{\alpha}, \boldsymbol{\theta}) = \sum_{n=1}^{N} \mathbb{E}_{q_{\boldsymbol{\alpha}}(d)} \left[ \log p(\mathbf{y}^{(n)}|\mathbf{x}^{(n)}, d; \boldsymbol{\theta}) \right] - \mathrm{KL}(q_{\boldsymbol{\alpha}}(d) \,\|\, p_{\boldsymbol{\beta}}(d)). \tag{2}$$

This Evidence Lower BOund (ELBO) allows us to optimise the variational parameters $\boldsymbol{\alpha}$ and network weights $\boldsymbol{\theta}$ simultaneously using gradients. Because both our variational and true posteriors are categorical, (2) is convex with respect to $\boldsymbol{\alpha}$. At the optima, $q_{\boldsymbol{\alpha}}(d) = p(d|\mathfrak{D}; \boldsymbol{\theta})$ and the bound is tight. Thus, we perform exact rather than approximate inference.

$\mathbb{E}_{q_{\boldsymbol{\alpha}}(d)}[\log p(\mathbf{y}|\mathbf{x}, d; \boldsymbol{\theta})]$ can be computed from the activations at every depth. Consequently, both terms in (2) can be evaluated exactly, with only a single forward pass. This removes the need for high variance Monte Carlo gradient estimators, often required by VI methods for NNs. When using mini-batches of size $B$, we stochastically estimate the ELBO in (2) as

$$\mathcal{L}(\boldsymbol{\alpha}, \boldsymbol{\theta}) \approx \frac{N}{B} \sum_{n=1}^{B} \sum_{i=0}^{D} \left( \log p(\mathbf{y}^{(n)}|\mathbf{x}^{(n)}, d{=}i; \boldsymbol{\theta}) \cdot \alpha_i \right) - \sum_{i=0}^{D} \left( \alpha_i \log \frac{\alpha_i}{\beta_i} \right). \tag{3}$$

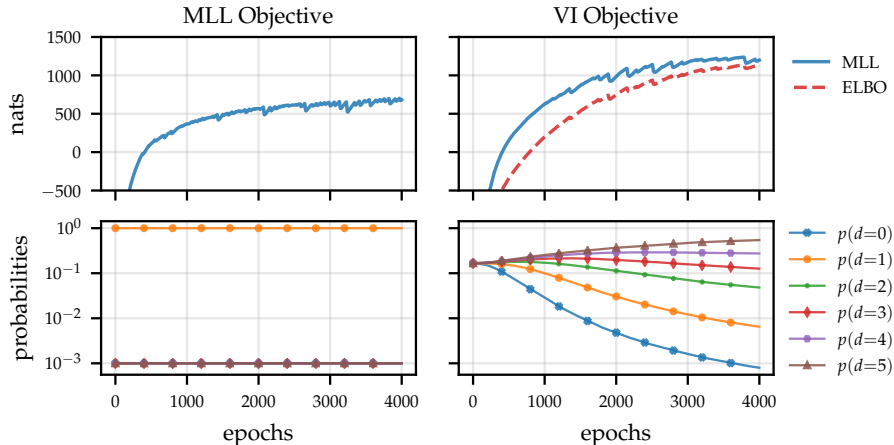

Figure 3: Top row: progression of MLL and ELBO during training. Bottom: progression of all six depth posterior probabilities. The left column corresponds to optimising the MLL directly and the right to VI. For the latter, variational posterior probabilities $q(d)$ are shown.

Predictions for new data $\mathbf{x}^*$ are made by marginalising depth with the variational posterior:

$$p(\mathbf{y}^*|\mathbf{x}^*, \mathfrak{D}; \boldsymbol{\theta}) = \sum_{i=0}^{D} p(\mathbf{y}^*|\mathbf{x}^*, d{=}i; \boldsymbol{\theta}) q_{\boldsymbol{\alpha}}(d{=}i). \tag{4}$$

## 4 Experiments

First, we compare the MLL and VI training approaches for DUNs. We then evaluate DUNs on toy-regression, real-world regression, and image classification tasks. As baselines, we provide results for vanilla NNs (denoted as 'SGD'), MC Dropout (Gal and Ghahramani, 2016), and deep ensembles (Lakshminarayanan et al., 2017), arguably the strongest approach for uncertainty estimation in deep learning (Snoek et al., 2019; Ashukha et al., 2020). For regression tasks, we also include Gaussian Mean Field VI (MFVI) (Blundell et al., 2015) with the local reparametrisation trick (Kingma et al., 2015). For the image classification tasks, we include stochastic depth ResNets (S-ResNets) (Huang et al., 2016), which can be viewed as MC Dropout applied to whole residual blocks, and deep ensembles of different depth networks (depth-ensembles). We include the former as an alternate method for converting uncertainty over depth into predictive uncertainty. We include the latter to investigate the hypothesis that the different classes of functions produced at different depths help provide improved disagreement and, in turn, predictive uncertainty. We study all methods in terms of accuracy, uncertainty quantification, and robustness to corrupted or OOD data. We place a uniform prior over DUN depth. See Appendix C, Appendix D, and Appendix E for detailed descriptions of the techniques we use to compute, and evaluate uncertainty estimates, and our experimental setup, respectively. Additionally, in Appendix G, we explore the effects of architecture hyperparameters on the depth posterior and the use of DUNs for architecture search. Code is available at `https://github.com/cambridge-mlg/DUN`.

### 4.1 Comparing MLL and VI training

Figure 3 compares the optimisation of a 5 hidden layer fully connected DUN on the concrete dataset using estimates of the MLL (1) and ELBO (3). The former approach converges to a local optima where all but one depth's probabilities go to 0. With VI, the surrogate posterior converges slower than the network weights. This allows $\boldsymbol{\theta}$ to reach a configuration where multiple depths can be used for prediction. Towards the end of training, the variational gap vanishes. The surrogate distribution approaches the true posterior without collapsing to a delta. The MLL values obtained with VI are larger than those obtained with (1), i.e. our proposed approach finds better explanations for the data. In Appendix B, we optimise (1) after reaching a local optima with VI (3). This does not cause posterior collapse, showing that MLL optimisation's poor performance is due to a propensity for poor local optima.

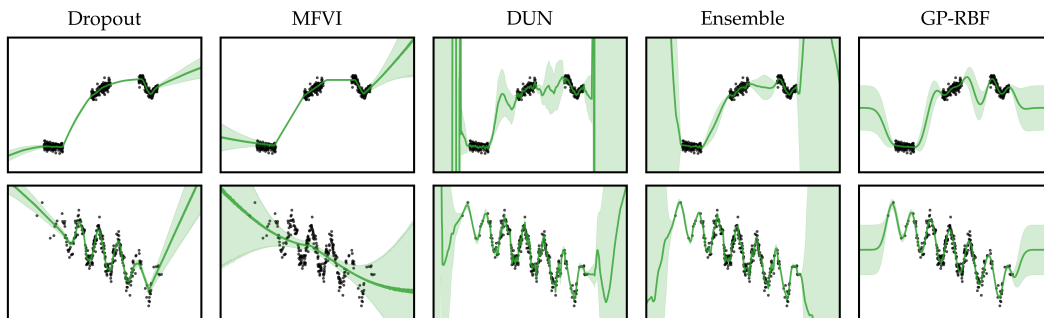

Figure 4: Top row: toy dataset from Izmailov et al. (2019). Bottom: Wiggle dataset. Black dots denote data points. Error bars represent standard deviation among mean predictions.

## 4.2 Toy Datasets

We consider two synthetic 1D datasets, shown in Figure 4. We use 3 hidden layer, 100 hidden unit, fully connected networks with residual connections for our baselines. DUNs use the same architecture but with 15 hidden layers. GPs use the RBF kernel. We found these configurations to work well empirically. In Appendix F.1, we perform experiments with different toy datasets, architectures and hyperparameters. DUNs' performance increases with depth but often 5 layers are sufficient to produce reasonable uncertainty estimates.

The first dataset, which is taken from Izmailov et al. (2019), contains three disjoint clusters of data. Both MFVI and Dropout present error bars that are similar in the data dense and in-between regions. MFVI underfits slightly, not capturing smoothness in the data. DUNs perform most similarly to Ensembles. They are both able to fit the data well and express in-between uncertainty. Their error bars become large very quickly in the extrapolation regime due to different ensemble elements' and depths' predictions diverging in different directions.

Our second dataset consists of 300 samples from $y = \sin(\pi x) + 0.2 \cos(4\pi x) - 0.3x + \epsilon$, where $\epsilon \sim \mathcal{N}(0, 0.25)$ and $x \sim \mathcal{N}(5, 2.5)$. We dub it "Wiggle". Dropout struggles to fit this faster varying function outside of the data-dense regions. MFVI fails completely. DUNs and Ensembles both fit the data well and provide error bars that grow as the data becomes sparse.

## 4.3 Tabular Regression

We evaluate all methods on UCI regression datasets using standard (Hernández-Lobato and Adams, 2015) and gap splits (Foong et al., 2019b). We also use the large-scale non-stationary flight delay dataset, preprocessed by Hensman et al. (2013). Following Deisenroth and Ng (2015), we train on the first 2M data points and test on the subsequent 100k. We select all hyperparameters, including NN depth, using Bayesian optimisation with HyperBand (Falkner et al., 2018). See Appendix E.2 for details. We evaluate methods with Root Mean Squared Error (RMSE), Log Likelihood (LL) and Tail Calibration Error (TCE). The latter measures the calibration of the 10% and 90% confidence intervals, and is described in Appendix D.

UCI standard split results are found in Figure 5. For each dataset and metric, we rank methods from 1 to 5 based on mean performance. We report mean ranks and standard deviations. Dropout obtains the best mean rank in terms of RMSE, followed closely by Ensembles. DUNs are third, significantly ahead of MFVI and SGD. Even so, DUNs outperform Dropout and Ensembles in terms of TCE, i.e. DUNs more reliably assign large error bars to points on which they make incorrect predictions. Consequently, in terms of LL, a metric which considers both uncertainty and accuracy, DUNs perform competitively (the LL rank distributions for all three methods overlap almost completely). MFVI provides the best calibrated uncertainty estimates. Despite this, its mean predictions are inaccurate, as evidenced by it being last in terms of RMSE. This leads to MFVI's LL rank only being better than SGD's. Results for gap splits, designed to evaluate methods' capacity to express in-between uncertainty, are given in Appendix F.2. Here, DUNs outperform Dropout in terms of LL rank. However, they are both outperformed by MFVI and ensembles.

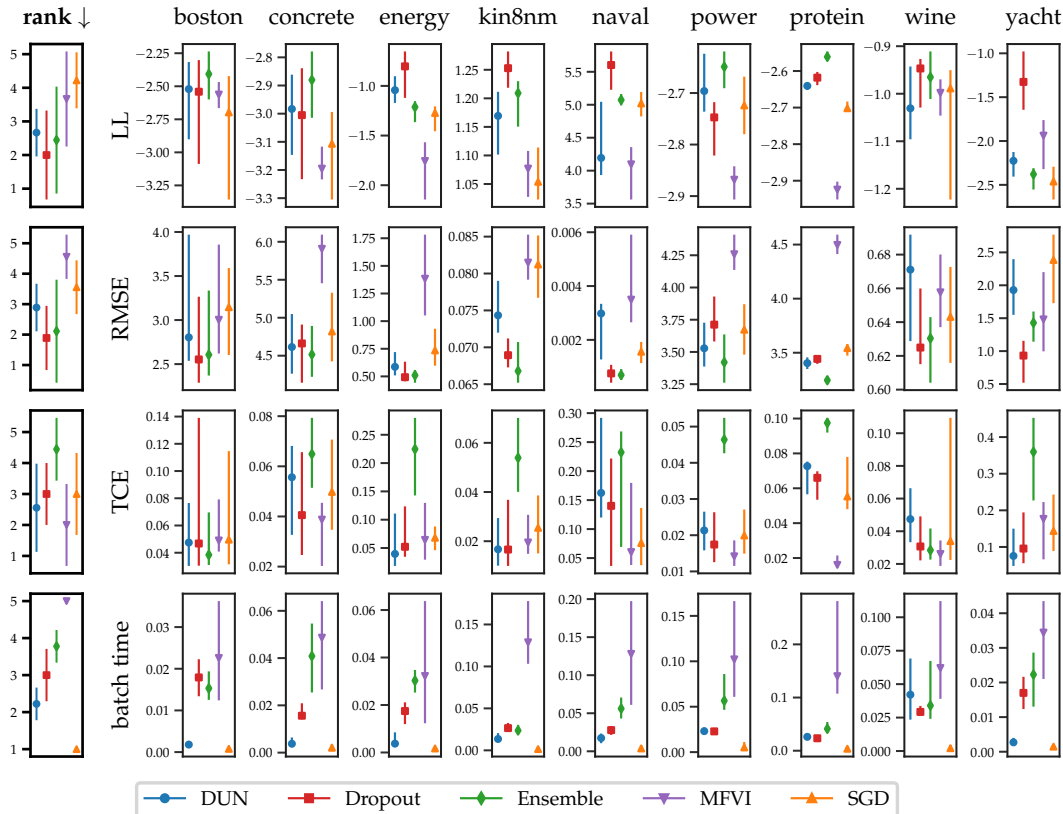

Figure 5: Quartiles for results on UCI regression datasets across standard splits. Average ranks are computed across datasets. For LL, higher is better. Otherwise, lower is better.

Table 1: Results obtained on the flights dataset (2M). Mean and standard deviation values are computed across 5 independent training runs.

| METRIC | DUN | DROPOUT | ENSEMBLE | MFVI | SGD |
|---|---|---|---|---|---|
| LL | $\mathbf{-4.95}_{\pm \mathbf{0.01}}$ | $\mathbf{-4.95}_{\pm \mathbf{0.02}}$ | $\mathbf{-4.95}_{\pm \mathbf{0.01}}$ | $-5.02_{\pm 0.05}$ | $-4.97_{\pm 0.01}$ |
| RMSE | $34.69_{\pm 0.28}$ | $\mathbf{34.28}_{\pm \mathbf{0.11}}$ | $34.32_{\pm 0.13}$ | $36.72_{\pm 1.84}$ | $34.61_{\pm 0.19}$ |
| TCE | $.087_{\pm .009}$ | $.096_{\pm .017}$ | $.090_{\pm .008}$ | $\mathbf{.068}_{\pm \mathbf{.014}}$ | $.084_{\pm .010}$ |
| Time | $.026_{\pm .001}$ | $.016_{\pm .001}$ | $.031_{\pm .001}$ | $.547_{\pm .003}$ | $\mathbf{.002}_{\pm \mathbf{.000}}$ |

The flights dataset is known for strong covariate shift between its train and test sets, which are sampled from contiguous time periods. LL values are strongly dependent on calibrated uncertainty. As shown in Table 1, DUNs' RMSE is similar to that of SGD, with Dropout and Ensembles performing best. Again, DUNs present superior uncertainty calibration. This allows them to achieve the best LL, tied with Ensembles and Dropout. We speculate that DUNs' calibration stems from being able to perform exact inference, albeit in depth space.

In terms of prediction time, DUNs clearly outrank Dropout, Ensembles, and MFVI on UCI. Due to depth, or maximum depth $D$ for DUNs, being chosen with Bayesian optimisation, methods' batch times vary across datasets. DUNs are often deeper because the quality of their uncertainty estimates improves with additional explanations of the data. As a result, SGD clearly outranks DUNs. On flights, increased depth causes DUNs' prediction time to lie in between Dropout's and Ensembles'.

## 4.4 Image Classification

We train ResNet-50 (He et al., 2016a) using all methods under consideration. This model is composed of an input convolutional block, 16 residual blocks and a linear layer. For DUNs, our prior over depth is uniform over the first 13 residual blocks. The last 3 residual blocks and linear layer form the output block, providing the flexibility to make predictions

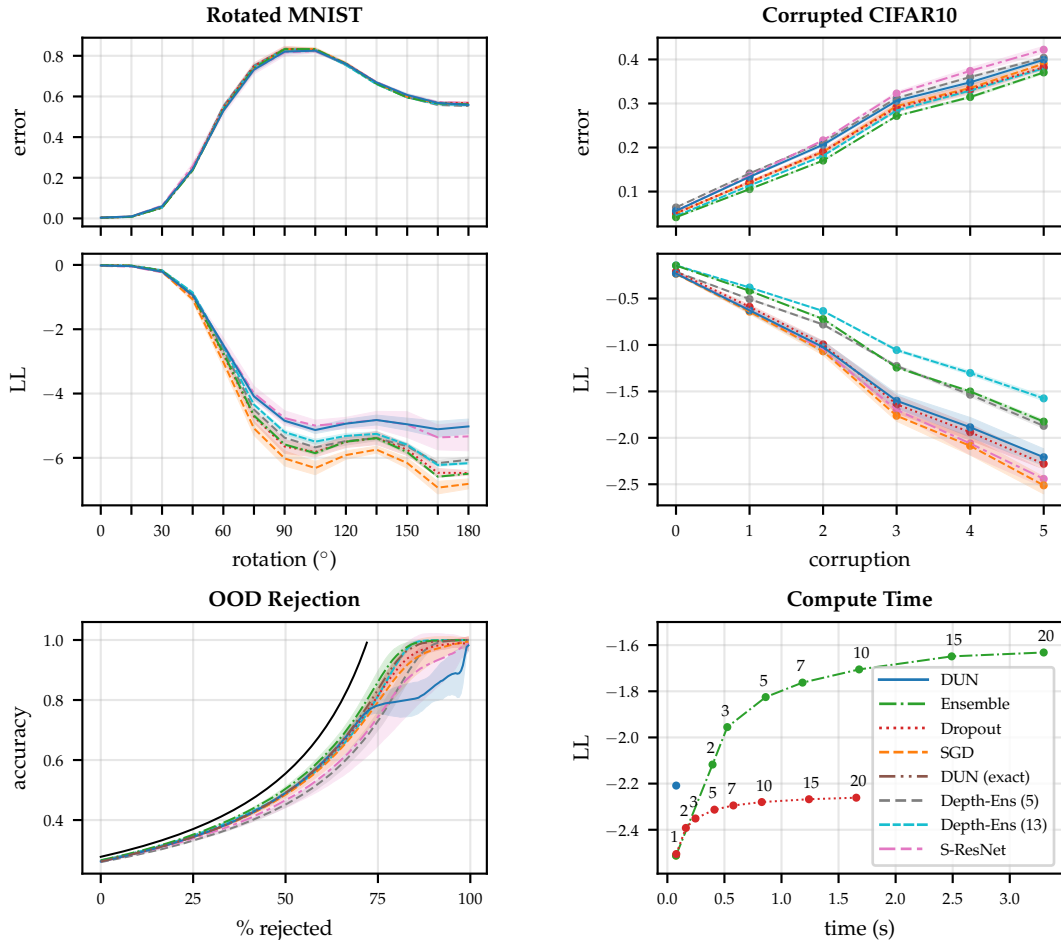

Figure 6: Top left: error and LL for MNIST at varying degrees of rotation. Top right: error and LL for CIFAR10 at varying corruption severities. Bottom left: CIFAR10-SVHN rejection-classification plot. The black line denotes the theoretical maximum performance; all in-distribution samples are correctly classified and OOD samples are rejected first. Bottom right: Pareto frontiers showing LL for corrupted CIFAR10 (severity 5) vs batch prediction time. Batch size is 256, split over 2 Nvidia P100 GPUs. Annotations show ensemble elements and Dropout samples. Note that a single element ensemble is equivalent to SGD.

from activations at multiple resolutions. We use $1 \times 1$ convolutions to adapt the number of channels between earlier blocks and the output block. We use default PyTorch training hyperparameters[2] for all methods. We set per-dataset LR schedules. We use 5 element (standard) deep ensembles, as suggested by Snoek et al. (2019), and 10 dropout samples. We use two variants of depth-ensembles. The first is composed of five elements corresponding to the five most shallow DUN sub-networks. The second depth-ensemble is composed of 13 elements, one for each depth used by DUNs. Similarly, S-ResNets are uncertain over only the first 13 layers. Figure 6 contains results for all experiments described below. Mean values and standard deviations are computed across 5 independent training runs. Full details are given in Appendix E.3.

**Rotated MNIST** Following Snoek et al. (2019), we train all methods on MNIST and evaluate their predictive distributions on increasingly rotated digits. Although all methods perform well on the original test-set, their accuracy degrades quickly for rotations larger than 30°. Here, DUNs and S-ResNets differentiate themselves by being the least overconfident. Additionally, depth-ensembles improve over standard ensembles. We hypothesize that predictions based on features at diverse resolutions allow for increased disagreement.

**Corrupted CIFAR** Again following Snoek et al. (2019), we train models on CIFAR10 and evaluate them on data subject to 16 different corruptions with 5 levels of intensity each (Hendrycks and Dietterich, 2019). Here, Ensembles significantly outperform all single network methods in terms of error and LL at all corruption levels, with depth-ensembles being notably better than standard ensembles. This is true even for the 5 element depth-ensemble, which has relatively shallow networks with far fewer parameters. The strong performance of depth-ensembles further validates our hypothesis that networks of different depths provide a useful diversity in predictions. With a single network, DUNs perform similarly to SGD and Dropout on the uncorrupted data. However, leveraging a distribution over depth allows DUNs to be the most robust non-ensemble method.

**OOD Rejection** We simulate a realistic OOD rejection scenario (Filos et al., 2019) by jointly evaluating our models on an in-distribution and an OOD test set. We allow our methods to reject increasing proportions of the data based on predictive entropy before classifying the rest. All predictions on OOD samples are treated as incorrect. Following Nalisnick et al. (2019b), we use CIFAR10 and SVHN as in and out of distribution datasets. Ensembles perform best. In their standard configuration, DUNs show underconfidence. They are incapable of separating very uncertain in-distribution inputs from OOD points. We re-run DUNs using the exact posterior over depth $p(d|\mathfrak{D};\boldsymbol{\theta})$ in (4), instead of $q_{\boldsymbol{\alpha}}(d)$. The exact posterior is computed while setting batch-norm to test mode. See Appendix F.3 for additional discussion. This resolves underconfidence, outperforming dropout and coming second, within error, of ensembles. We don't find exact posteriors to improve performance in any other experiments. Hence we abstain from using them, as they require an additional evaluation of the train set.

**Compute Time** We compare methods' performance on corrupted CIFAR10 (severity 5) as a function of computational budget. The LL obtained by a DUN matches that of a ~1.8 element ensemble. A single DUN forward pass is ~1.02 times slower than a vanilla network's. On average, DUNs' computational budget matches that of ~0.47 ensemble elements or ~0.94 dropout samples. These values are smaller than one due to overhead such as ensemble element loading. Thus, making predictions with DUNs is 10× faster than with five element ensembles. Note that we include loading times for ensembles to reflect that it is often impractical to store multiple ensemble elements in memory. Without loading times, ensemble timing would match Dropout. For *single-element ensembles* (SGD) we report only the prediction time.

## 5  Discussion and Future Work

We have re-cast NN depth as a random variable, rather than a fixed parameter. This treatment allows us to optimise weights as model hyperparameters, preserving much of the simplicity of non-Bayesian NNs. Critically, both the model evidence and predictive posterior for DUNs can be evaluated with a single forward pass. Our experiments show that networks of different depths obtain diverse fits. As a result, DUNs produce well calibrated uncertainty estimates, performing well relative to their computational budget on uncertainty-aware tasks. They scale to modern architectures and large datasets.

In DUNs, network weights have dual roles: fitting the data well and expressing diverse predictive functions at each depth. In future work, we would like to develop optimisation schemes that better ensure both roles are fulfilled, and investigate the relationship between excess model capacity and DUN performance. We would also like to investigate the effects of DUN depth on uncertainty estimation, allowing for more principled model selection. Additionally, because depth uncertainty is orthogonal to weight uncertainty, both could potentially be combined to expand the space of hypothesis over which we perform inference. Furthermore, it would be interesting to investigate the application of DUNs to a wider range of NN architectures, for example stacked RNNs or Transformers.

## Broader Impact

We have introduced a general method for training neural networks to capture model uncertainty. These models are fairly flexible and can be applied to a large number of applications, including potentially malicious ones. Perhaps, our method could have the largest impact on critical decision making applications, where reliable uncertainty estimates are as important as the predictions themselves. Financial default prediction and medical diagnosis would be examples of these.

We hope that this work will contribute to increased usage of uncertainty aware deep learning methods in production. DUNs are trained with default hyperparameters and easy to make converge to reasonable solutions. The computational cost of inference in DUNs is similar to that of vanilla NNs. This makes DUNs especially well suited for applications with real-time requirements or low computational resources, such as self driving cars or sensor fusion on embedded devices. More generally, DUNs make leveraging uncertainty estimates in deep learning more accessible for researchers or practitioners who lack extravagant computational resources.

Despite the above, a hypothetical failure of our method, e.g. providing miscalibrated uncertainty estimates, could have large negative consequences. This is particularly the case for critical decision making applications, such as medical diagnosis.

## Acknowledgments and Disclosure of Funding

We would like to thank Eric Nalisnick and John Bronskill for helpful discussions. We also thank Pablo Morales-Álvarez, Stephan Gouws, Ulrich Paquet, Devin Taylor, Shakir Mohamed, Avishkar Bhoopchand and Taliesin Beynon for giving us feedback on this work. Finally, we thank Marc Deisenroth and Balaji Lakshminarayanan for helping us acquire the flights dataset and Andrew Foong for providing us with the UCI gap datasets.

JA acknowledges support from Microsoft Research, through its PhD Scholarship Programme, and from the EPSRC. JUA acknowledges funding from the EPSRC and the Michael E. Fisher Studentship in Machine Learning. This work has been performed using resources provided by the Cambridge Tier-2 system operated by the University of Cambridge Research Computing Service (http://www.hpc.cam.ac.uk) funded by EPSRC Tier-2 capital grant EP/P020259/1.

## Footnotes

[2]https://github.com/pytorch/examples/blob/master/imagenet/main.py

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
