[Supplementary Material]

## Appendix

This appendix is arranged as follows:

- We derive the lower bound used to train DUNs in Appendix A.
- We analyse the proposed MLE (1) and VI (2) objectives in Appendix B.
- We discuss how to compute uncertainty estimates with all methods under consideration in Appendix C.
- We discuss approaches to evaluate the quality of uncertainty estimates in Appendix D.
- We detail the experimental setup used for training and evaluation in Appendix E.
- We provide additional experimental results in Appendix F.
- We discuss the application of DUNs to neural architecture search in Appendix G.
- We show how standard PyTorch NNs can be adapted into DUNs in Appendix H.
- We provide some negative results in Appendix I.

## A  Derivation of (2) and link to the EM algorithm

Referring to $\mathfrak{D}=\{\mathbf{X}, \mathbf{Y}\}$ with $\mathbf{X} = \{\mathbf{x}^{(n)}\}_{n=1}^N$, and $\mathbf{Y} = \{\mathbf{y}^{(n)}\}_{n=1}^N$, we show that (2) is a lower bound on $\log p(\mathfrak{D}; \boldsymbol{\theta}) = \log p(\mathbf{Y}|\mathbf{X}; \boldsymbol{\theta})$:

$$
\begin{aligned}
\mathrm{KL}(q_{\boldsymbol{\alpha}}(d) \,\|\, p(d|\mathfrak{D}; \boldsymbol{\theta})) &= \mathbb{E}_{q_{\boldsymbol{\alpha}}(d)}[\log q_{\boldsymbol{\alpha}}(d) - \log p(d|\mathfrak{D})] \\
&= \mathbb{E}_{q_{\boldsymbol{\alpha}}(d)}\left[\log q_{\boldsymbol{\alpha}}(d) - \log \frac{p(\mathbf{Y}|\mathbf{X}, d; \boldsymbol{\theta})\, p(d)}{p(\mathbf{Y}|\mathbf{X}; \boldsymbol{\theta})}\right] \\
&= \mathbb{E}_{q_{\boldsymbol{\alpha}}(d)}[\log q_{\boldsymbol{\alpha}}(d) - \log p(\mathbf{Y}|\mathbf{X}, d; \boldsymbol{\theta}) - \log p(d) + \log p(\mathbf{Y}|\mathbf{X}; \boldsymbol{\theta})] \\
&= \mathbb{E}_{q_{\boldsymbol{\alpha}}(d)}[-\log p(\mathbf{Y}|\mathbf{X}, d; \boldsymbol{\theta})] + \mathrm{KL}(q_{\boldsymbol{\alpha}}(d) \,\|\, p(d)) + \log p(\mathbf{Y}|\mathbf{X}; \boldsymbol{\theta}) \\
&= -\mathcal{L}(\boldsymbol{\alpha}, \boldsymbol{\theta}) + \log p(\mathbf{Y}|\mathbf{X}; \boldsymbol{\theta}).
\end{aligned}
\tag{5}
$$

Using the non-negativity of the KL divergence, we can see that: $\mathcal{L}(\boldsymbol{\alpha}, \boldsymbol{\theta}) \leq \log p(\mathbf{Y}|\mathbf{X}; \boldsymbol{\theta})$.

We now discuss the link to the EM algorithm introduced in Section 3.2. Recall that, in our model, network depth $d$ acts as the latent variable and network weights $\boldsymbol{\theta}$ are parameters. For a given setting of network weights $\boldsymbol{\theta}^k$, at optimisation step $k$, we can apply Bayes rule to perform the *E step*, obtaining the exact posterior over $d$:

$$
\alpha_j^{k+1} = p(d=j|\mathfrak{D}; \boldsymbol{\theta}^k) = \frac{p(d=j) \cdot \prod_{n=1}^N p(\mathbf{y}^{(n)}|\mathbf{x}^{(n)}, d=j; \boldsymbol{\theta}^k)}{\sum_{i=0}^D p(d=i) \cdot \prod_{n=1}^N p(\mathbf{y}^{(n)}|\mathbf{x}^{(n)}, d=i; \boldsymbol{\theta}^k)}
\tag{6}
$$

The posterior depth probabilities can now be used to marginalise this latent variable and perform maximum likelihood estimation of network parameters. This is the *M step*:

$$
\begin{aligned}
\boldsymbol{\theta}^{k+1} &= \arg\max_{\boldsymbol{\theta}} \mathbb{E}_{p(d|\mathfrak{D}; \boldsymbol{\theta}^k)}\left[\prod_{n=1}^N p(\mathbf{y}^{(n)}|\mathbf{x}^{(n)}, d; \boldsymbol{\theta}^k)\right] \\
&= \arg\max_{\boldsymbol{\theta}} \sum_{i=0}^D p(d=i|\mathfrak{D}; \boldsymbol{\theta}^k) \prod_{n=1}^N p(\mathbf{y}^{(n)}|\mathbf{x}^{(n)}, d=i; \boldsymbol{\theta}^k)
\end{aligned}
\tag{7}
$$

The E step (6) requires calculating the likelihood of the complete training dataset. The M step requires optimising the weights of the NN. Both operations are expensive when dealing with large networks and big data. The EM algorithm is not practical in this case, as requires performing both steps multiple times. We sidestep this issue through the introduction of an approximate posterior $q(d)$, parametrised by $\boldsymbol{\alpha}$, and a variational lower bound on the marginal log-likelihood (5). The corresponding variational E step is given by:

$$
\boldsymbol{\alpha}^{k+1} = \arg\max_{\boldsymbol{\alpha}} \sum_{n=1}^N \mathbb{E}_{q_{\boldsymbol{\alpha}}(d)}\left[\log p(\mathbf{y}^{(n)}|\mathbf{x}^{(n)}, d; \boldsymbol{\theta}^k)\right] - \mathrm{KL}(q_{\boldsymbol{\alpha}^{(k)}}(d) \,\|\, p_{\boldsymbol{\beta}}(d))
\tag{8}
$$

Because our variational family contains the exact posterior distribution – they are both categorical – the ELBO is tight at the optima with respect to the variational parameters $\boldsymbol{\alpha}$. Solving (8) recovers $\boldsymbol{\alpha}$ such that $q_{\boldsymbol{\alpha}^{k+1}}(d) = p(d|\mathfrak{D}; \boldsymbol{\theta}^k)$. This step can be performed with stochastic gradient optimisation.

We can now combine the variational E step (8) and M step (7) updates, recovering (2), where $\boldsymbol{\alpha}$ and $\boldsymbol{\theta}$ are updated simultaneously through gradient steps:

$$\mathcal{L}(\boldsymbol{\alpha}, \boldsymbol{\theta}) = \sum_{n=1}^{N} \mathbb{E}_{q_{\boldsymbol{\alpha}}(d)} \left[ \log p(\mathbf{y}^{(n)}|\mathbf{x}^{(n)}, d; \boldsymbol{\theta}) \right] - \mathrm{KL}(q_{\boldsymbol{\alpha}}(d) \,\|\, p(d))$$

This objective is amenable to minibatching. The variational posterior tracks the true posterior during gradient updates. Thus, (2), allows us to optimise a lower bound on the data's marginal log-likelihood which is unbiased in the limit.

## B  Comparing VI and MLL Training Objectives

In this section, we further compare the MLL (1) and VI (3) training objectives presented in Section 3.2. Our probabilistic model is atypical in that it can have millions of hyperparameters, NN weights, while having a single latent variable, depth. For moderate to large datasets, the posterior distribution over depth is determined almost completely by the setting of the network weights. The success of DUNs is largely dependent on being able to optimise these hyperparameters well. Even so, our probabilistic model tells us nothing about how to do this. We investigate the gradients of both objectives with respect to the hyperparameters. For MLL:

$$\frac{\partial}{\partial \boldsymbol{\theta}} \log p(\mathfrak{D}; \boldsymbol{\theta}) = \frac{\partial}{\partial \boldsymbol{\theta}} \mathrm{logsumexp}_d (\log p(\mathfrak{D}|d; \boldsymbol{\theta}) + \log p(d))$$

$$= \sum_{i=0}^{D} \frac{p(\mathfrak{D}|d=i; \boldsymbol{\theta})p(d=i)}{\sum_{j=0}^{D} p(\mathfrak{D}|d=j; \boldsymbol{\theta})p(d=j)} \frac{\partial}{\partial \boldsymbol{\theta}} \log p(\mathfrak{D}|d=i; \boldsymbol{\theta})$$

$$= \sum_{i=0}^{D} p(d=i|\mathfrak{D}; \boldsymbol{\theta}) \frac{\partial}{\partial \boldsymbol{\theta}} \log p(\mathfrak{D}|d=i; \boldsymbol{\theta})$$

$$= \mathbb{E}_{p(d|\mathfrak{D}; \boldsymbol{\theta})} [\frac{\partial}{\partial \boldsymbol{\theta}} \log p(\mathfrak{D}|d; \boldsymbol{\theta})] \qquad (9)$$

The gradient of the marginal log-likelihood is equivalent to expectation, under the posterior over depth, of the gradient of the log-likelihood conditioned on depth. The weights of the subnetwork which is able to best explain the data at initialisation will receive larger gradients. This will result in this depth fitting the data even better and receiving larger gradients in successive iterations while the gradients for subnetworks of different depths vanish, i.e. the rich get richer. We conjecture that the MLL objective is prone to hard-to-escape local optima, at which a single depth is used. This can be especially problematic if the initial posterior distribution has its maximum over shallow depths, as this will reduce the capacity of the NN.

On the other hand, VI decouples the likelihood at each depth from the approximate posterior during optimisation:

$$\frac{\partial}{\partial \boldsymbol{\theta}} \mathcal{L}(\boldsymbol{\theta}, \boldsymbol{\alpha}) = \sum_{i=0}^{D} q_{\boldsymbol{\alpha}}(d=i) \frac{\partial}{\partial \boldsymbol{\theta}} \log p(\mathfrak{D}|d=i; \boldsymbol{\theta})$$

$$\frac{\partial}{\partial \alpha_i} \mathcal{L}(\boldsymbol{\theta}, \boldsymbol{\alpha}) = \underbrace{\log p(\mathfrak{D}|d=i; \boldsymbol{\theta}) \frac{\partial}{\partial \alpha_i} q_{\boldsymbol{\alpha}}(d=i)}_{\mathrm{I}} - (\log q_{\boldsymbol{\alpha}}(d=i) - \log p(d=i) + 1) \frac{\partial}{\partial \alpha_i} q_{\boldsymbol{\alpha}}(d=i)$$

$$(10)$$

For moderate to large datasets, when updating the variational parameters $\boldsymbol{\alpha}$, the data dependent term (I) of the ELBO's gradient will dominate. However, the gradients that reach the variational parameters are scaled by the log-likelihood at each depth. In contrast, in (9), the likelihood at each depth scales the gradients directly. We conjecture that, with VI, $\boldsymbol{\alpha}$

Figure 7: Top row: progression of the MLL and ELBO during training of ResNet-50 DUNs on the Fashion dataset. Bottom: progression of depth posterior probabilities. The left column corresponds to MLL optimisation and the right to VI. For the latter, approximate posterior probabilities are shown. We perform an additional 10 epochs of "finetunning" on the VI DUN with the MLL objective. These are separated by the vertical black line. True posterior probabilities are shown for these 10 epochs. The posterior over depth, ELBO and MLL values shown are not stochastic estimates. They are computed using the full training set.

will converge slower than the true posterior does when optimising the MLL directly. This allows network weights to reach to solutions that explain the data well at multiple depths.

We test the above hypothesis by training a ResNet-50 DUN on the Fashion-MNIST dataset, as shown in Figure 7. We treat the first 7 residual blocks of the model as the DUNs input block and the last 3 as the output block. This leaves us with the need to infer a distribution over 5 depths (7-12). Both the MLL and VI training schemes run for 90 epochs, with scheduling described in Appendix E.3. We then fine-tune the DUN that was trained with VI for 10 additional epochs using the MLL objective. Both training schemes obtain very similar MLL values. The dataset under consideration is much larger than the one in Section 4.1, but the dimensionality of the latent variable stays the same. Hence, the variational gap is small relative to the MLL. Nevertheless, unlike with the MLL objective, VI training results in posteriors that avoid placing all of their mass on a single depth setting.

Figure 8: Zoomed-in view of the last 20 epochs of Figure 7. The vertical black line denotes the switch from VI training to MLL optimisation. Probabilities to the left of the line correspond to the variational posterior $q$. The ones to the right of the line correspond to the exact posterior. In some steps of training, the ELBO appears to be larger than the MLL due to numerical error.

Zooming in on the last 20 epochs in Figure 8, we see that after converging to a favorable solution with VI, optimising the MLL objective directly does not result in the posterior collapsing to a single depth. Instead, it remains largely the same as the VI surrogate posterior. VI optimisation allowed us to find an optima of the MLL where multiple depths explain the data similarly well.

In Figure 9 and Figure 10 we show the MLL, ELBO and posterior probabilities obtained with our two optimisation objectives, (1) and (3), on the Boston and Wine datasets respectively. Like in Section 4.1, we employ 5 hidden layer DUNs without residual connections and 100 hidden units per layer. The input and output blocks consist of linear layers. Both approaches employ full-batch gradient descent with a step-size of $10^{-3}$ and momentum of 0.9.

Figure 9: Top row: progression of MLL and ELBO during training of DUNs on the Boston dataset. Bottom: progression of depth posterior probabilities. The left column corresponds to MLL optimisation and the right to VI. For the latter, approximate posterior probabilities are shown.

Figure 10: Top row: progression of MLL and ELBO during training of DUNs on the Wine dataset. Bottom: progression of depth posterior probabilities. The left column corresponds to MLL optimisation and the right to VI. For the latter, approximate posterior probabilities are shown.

The MLL objective consistently reaches parameter settings for which the posterior over depth places all its mass on a single depth. We found the depth to which the posterior collapses to change depending on weight initialisation. However, converging to a network where no hidden layers were used $p(d{=}0){=}1$ seems to be the most common occurrence. Even when the chosen depth is large, as in the Wine dataset example, we are able to reach significantly larger likelihood values when optimising the ELBO. Even though the variational gap becomes very small by the end of training, the approximate posterior probabilities found with VI place non-0 mass over more than one depth; training with VI allows us to find weight configurations which explain the data well while being able to use multiple layers for prediction.

## C    Computing Uncertainties

In this work, we consider NNs which parametrise two types of distributions over target variables: the categorical for classification problems and the Gaussian for regression. For generality, in this section we omit references to model hyperparameters $\boldsymbol{\theta}$ and refer to the distribution over random variables that induces stochasticity in our networks as $q(\mathbf{w})$. In DUNs, this is a distribution over depth. It is a distribution over weights in the case of MFVI, MC Dropout and ensembles.

For classification models, our networks output a probability vector with elements $f_k(\mathbf{x}, \mathbf{w})$, corresponding to classes $\{c_k\}_{k=1}^K$. The likelihood function is $p(y|\mathbf{x}, \mathbf{w}) = \text{Cat}(y; f(\mathbf{x}, \mathbf{w}))$. Through marginalisation, the uncertainty in $\mathbf{w}$ is translated into uncertainty in predictions. For DUNs, computing the exact predictive posterior is tractable (4). However, for our baseline approaches, we resort to approximating it with $M$ MC samples:

$$p(\mathbf{y}^*|\mathbf{x}^*, \mathfrak{D}) = \mathbb{E}_{p(\mathbf{w}|\mathfrak{D})}[p(\mathbf{y}^*|\mathbf{x}^*, \mathbf{w})]$$

$$\approx \frac{1}{M} \sum_{m=0}^M f(\mathbf{x}^*, \mathbf{w}); \quad \mathbf{w} \sim q(\mathbf{w})$$

In both, the exact and approximate cases, the resulting predictive distribution is categorical. We quantify its uncertainty using entropy:

$$H(\mathbf{y}^*|\mathbf{x}^*, \mathfrak{D}) = \sum_{k=1}^K p(y^* = c_k|\mathbf{x}^*, \mathfrak{D}) \log p(y^* = c_k|\mathbf{x}^*, \mathfrak{D})$$

For regression, we employ homoscedastic likelihood functions. The mean is parametrised by a NN and the variance is learnt as a standalone parameter: $p(\mathbf{y}^*|\mathbf{x}^*, \mathbf{w}) = \mathcal{N}(\mathbf{y}; f(\mathbf{x}^*, \mathbf{w}), \boldsymbol{\sigma}^2 \cdot I)$. For the models under consideration, marginalising over $\mathbf{w}$ induces a Gaussian mixture distribution over outputs. We approximate this mixture with a single Gaussian using moment matching: $p(\mathbf{y}^*|\mathbf{x}^*) \approx \mathcal{N}(\mathbf{y}; \boldsymbol{\mu}_a, \boldsymbol{\sigma}_a^2)$. For DUNs, the mean can be computed exactly:

$$\boldsymbol{\mu}_a = \sum_{i=0}^D f(\mathbf{x}^*, \mathbf{w} = i) q(\mathbf{w} = i)$$

Otherwise, we estimate it with MC:

$$\boldsymbol{\mu}_a \approx \frac{1}{M} \sum_{m=0}^M f(\mathbf{x}^*, \mathbf{w}); \quad \mathbf{w} \sim q(\mathbf{w})$$

The predictive variance is obtained as the variance of the GMM. For DUNs:

$$\boldsymbol{\sigma}_a^2 = \underbrace{\sum_{i=0}^D q(\mathbf{w} = i) f(\mathbf{x}^*, \mathbf{w} = i)^2 - \boldsymbol{\mu}_a^2}_{\text{I}} + \underbrace{\boldsymbol{\sigma}^2}_{\text{II}};$$

Otherwise, we estimate it with MC:

$$\boldsymbol{\sigma}_a^2 \approx \underbrace{\frac{1}{M} \sum_{m=1}^M f(\mathbf{x}^*, \mathbf{w})^2 - \boldsymbol{\mu}_a^2}_{\text{I}} + \underbrace{\boldsymbol{\sigma}^2}_{\text{II}}; \quad \mathbf{w} \sim q(\mathbf{w})$$

Here, I reflects model uncertainty – our lack of knowledge about $\mathbf{w}$ – while II tells us about the irreducible uncertainty or noise in our training data.

## D    Evaluating Uncertainty Estimates

We consider the following approaches to quantify the quality of uncertainty estimates:

- **Test Log Likelihood** (higher is better): This metric tells us about how probable it is that the test targets where generated using the test inputs and our model. It is a proper scoring rule (Gneiting and Raftery, 2007) that depends on both the accuracy of predictions and their uncertainty. We employ it in both classification and regression settings, using categorical and Gaussian likelihoods, respectively.

- **Brier Score** (lower is better): Proper scoring rule that measures the accuracy of predictive probabilities in classification tasks. It is computed as the mean squared distance between predicted class probabilities and one-hot class labels:

$$\text{BS} = \frac{1}{N} \sum_{n=1}^{N} \frac{1}{K} \sum_{k=1}^{K} (p(y^* = c_k | \mathbf{x}^*, \mathfrak{D}) - \mathbb{1}[y^* = c_k])^2$$

Erroneous predictions made with high confidence are penalised less by Brier score than by log-likelihood. This can avoid outlier inputs from having a dominant effect on experimental results. Nevertheless, we find Brier score to be less sensitive than log-likelihood, making it harder to distinguish the approaches being compared. Hence, we favor the use of log-likelihood in Section 4.4.

- **Expected Calibration Error (ECE)** (lower is better): This metric measures the difference between predictive confidence and empirical accuracy in classification. It is computed by dividing the [0,1] range into a set of bins $\{B_s\}_{s=1}^{S}$ and weighing the miscalibration in each bin by the number of points that fall into it $|B_s|$:

$$\text{ECE} = \sum_{s=1}^{S} \frac{|B_s|}{N} |\text{acc}(B_s) - \text{conf}(B_s))|$$

Here,

$$\text{acc}(B_s) = \frac{1}{|B_s|} \sum_{\mathbf{x} \in B_s} \mathbb{1}[\mathbf{y} = \arg\max_{c_k} p(\mathbf{y}|\mathbf{x}, \mathfrak{D})] \quad \text{and}$$

$$\text{conf}(B_s) = \frac{1}{|B_s|} \sum_{\mathbf{x} \in B_s} \max p(\mathbf{y}|\mathbf{x}, \mathfrak{D}).$$

ECE is not a proper scoring rule. A perfect ECE score can be obtained by predicting the marginal distribution of class labels $p(\mathbf{y})$ for every input. A well calibrated predictor with poor accuracy would obtain low log likelihood values but also low ECE. Although ECE works well for binary classification, the naive adaption to the multi-class setting suffers from a number of pathologies (Nixon et al., 2019). Thus, we do not employ this metric.

- **Regression Calibration Error (RCE)** (lower is better): We extend ECE to regression settings, while avoiding the pathologies described by Nixon et al. (2019): We seek to asses how well our model's predictive distribution describes the residuals obtained on the test set. It is not straight forward to define bins, like in standard ECE, because our predictive distribution might not have finite support. We apply the CDF of our predictive distribution to our test targets. If the predictive distribution describes the targets well, the transformed distribution should resemble a uniform with support [0, 1]. This procedure is common for backtesting market risk models (Dowd, 2013).

To asses the global similarity between our targets' distribution and our predictive distribution, we separate the [0, 1] interval into $S$ equal-sized bins $\{B_s\}_{s=1}^{S}$. We compute calibration error in each bin as the difference between the proportion of points that have fallen within that bin and $1/s$:

$$\text{RCE} = \sum_{s=1}^{S} \frac{|B_s|}{N} \cdot |\frac{1}{S} - \frac{|B_s|}{N}|; \quad |B_s| = \sum_{n=1}^{N} \mathbb{1}[CDF_{p(y|\mathbf{x}^{(n)})}(y^{(n)}) \in B_s]$$

Alternatively, we can asses how well our model predicts extreme values with a "frequency of tail losses" approach (Kupiec, 1995). It might not be realistic to assume

the noise in our targets is Gaussian. Only considering calibration at the tails of the predictive distribution allows us to ignore shape mismatch between the predictive distribution and the true distribution over targets. Instead, we focus on our model's capacity to predict on which inputs it is likely to make large mistakes. This can be used to ensure our model is not overconfident OOD. We specify two bins $\{B_0, B_1\}$, one at each tail end of our predictive distribution, and compute **Tail Calibration Error (TCE)** as:

$$\text{TCE} = \sum_{s=0}^{1} \frac{|B_s|}{|B_0| + |B_1|} \cdot \left| \frac{1}{\tau} - \frac{|B_s|}{N} \right|;$$

$$|B_0| = \sum_{n=1}^{N} \mathbb{1}[CDF_{p(y|\mathbf{x}^{(n)})}(y^{(n)}) < \tau]; \quad |B_1| = \sum_{n=1}^{N} \mathbb{1}[CDF_{p(y|\mathbf{x}^{(n)})}(y^{(n)}) \geq (1-\tau)]$$

We specify the tail range of our distribution by selecting $\tau$. Note that this is slightly different from Kupiec (1995), who uses a binomial test to asses whether a model's predictive distribution agrees with the distribution over targets in the tails.

RCE and TCE are not a proper scoring rules. Additionally, they are only applicable to 1 dimensional continuous target variables.

Please see (Ashukha et al., 2020; Snoek et al., 2019) for additional discussion on evaluating uncertainty estimates of predictive models.

# E   Experimental Setup

We implement all of our experiments in PyTorch (Paszke et al., 2019). Gaussian processes for toy data experiments are implemented with GPyTorch (Gardner et al., 2018).

## E.1   Toy Dataset Experiments

All NNs used for toy regression experiments in Section 4.2 consist of fully connected models with ReLU activations and residual connections. Their hidden layer width is 100. Batch normalisation is applied after every layer for SGD and DUNs. Unless specified otherwise, the same is true for the additional toy dataset experiments conducted in Appendix F.1. Network depths are defined on a per-experiment basis. DUNs employ linear input and output blocks, meaning that a depth of $d=0$ corresponds to a linear model. We refer to depth as the number of hidden layers of a NN.

Ensemble elements, DUNs and dropout models employ a weight decay value of $10^{-4}$. Ensembles are composed of 20 identical networks, trained from different initialisations. Initialisation parameters are sampled from the He initialisation (He et al., 2015). Dropout probabilities are fixed to 0.1. MFVI networks use a $\mathcal{N}(\mathbf{0}, I)$ prior. Gradients of the likelihood term in the ELBO are estimated with the local reparameterisation trick (Kingma et al., 2015) using 5 MC samples. DUNs employ uniform priors, assigning the same mass to each depth.

Networks are optimised using 6000 steps of full-batch gradient descent with a momentum value of 0.9 and learning rate of $10^{-3}$. Exceptions to this are: Dropout being trained for 10000 epochs, as we found 6000 to not be enough to achieve convergence, and MFVI using a learning rate of $10^{-2}$. For MFVI and DUNs, we scale the ELBO by one over the number of data points $N$. This makes the scale of the objective insensitive to dataset size.

The parameters of the predictive distributions are computed as described in Appendix C. For 1D datasets, we draw $10^4$ MC samples with MFVI and dropout. For 2D datasets, we draw $10^3$. Plot error bars correspond to the standard deviations of each approach's mean predictions. Thus, they convey model uncertainty.

Gaussian processes use a Gaussian likelihood function and radial basis function (RBF) kernel. For 2d toy experiments, the automatic relevance detection (ARD) version of the kernel is used, allowing for a different length-scale per dimension. A gamma prior with parameters

$\alpha = 1, \beta = 20$ is placed on the length-scale parameter for the 1d datasets. This avoids local optima of the log-likelihood function where fast varying patterns in the data are treated like noise. Noise variance and kernel parameters are learnt by optimising the MLL with 100 steps of Adam. Step size is set to 0.1.

We employ 7 different toy datasets. These allow us to test methods' capacity to express uncertainty in-between clusters of data and outside the convex hull of the data. They also allow us to evaluate methods' capacity to fit differently quickly varying functions. All of them can be loaded using our provided code.

### E.2 Regression Experiments

### E.2.1 Hyperparameter Optimisation and Training

To obtain our results on tabular regression tasks, given in Section 4.3, we follow Hernández-Lobato and Adams (2015) and follow-up work (Gal and Ghahramani, 2016; Lakshminarayanan et al., 2017) in performing Hyperparameter Optimisation (HPO) to determine the best configurations for each method. However, rather than using Bayesian Optimisation (BO) (Snoek et al., 2012) we use Bayesian Optimisation and Hyperband (BOHB) (Falkner et al., 2018). This method, as the name suggests, combines BO with Hyperband, a bandit based HPO method (Li et al., 2017). BOHB has the strengths of both BO (strong final performance) and Hyperband (scalability, flexibility, and robustness).

In particular, we use the `HpBandSter` implementation of BOHB: `https://github.com/automl/HpBandSter`. We run BOHB for each dataset and split for 20 iterations using the same settings, shown in Table 2. `min_budget` and `max_budget` are defined on a per dataset basis, as shown in Table 3. We find these values to be sufficiently large to ensure all methods' convergence.

Table 2: BOHB settings.

| SETTING | VALUE |
|---|---|
| eta | 3 |
| min_points_in_model | None |
| top_n_percent | 14 |
| num_samples | 64 |
| random_fraction | 1/3 |
| bandwidth_factor | 3 |
| min_bandwidth | 1e-3 |

For each test-train split of each dataset, we split the original training set into a new training set and a validation set. The validation sets are taken to be the last $N$ elements of the original training set, where $N$ is calculated from the validation proportions listed in Table 3. The training and validation sets are normalised by subtracting the mean and dividing by the variance of the new training set. BOHB performs minimisation on the validation Negative Log Likelihood (NLL). During optimisation, we perform early stopping with patience values show in Table 3.

As shown in Table 4, each method has a different set of hyperparameters to optimise. The BOHB configuration for each hyperparameter is shown in Table 5. It is worth noting that maximum network depth is a hyperparater which we optimise with BOHB. DUNs benefit from being deeper as it allows then to perform BMA over a larger set of functions. We prevent this from disadvantaging competing methods by choosing the depth at which each one performs best.

All methods are applied to fully-connected networks with hidden layer width of 100. We employ residual connections, allowing all approaches to better take advantage of depth. All methods are trained using SGD with momentum and a batch size of 128. No learning rate scheduling is performed. We use batch-normalisation for DUNs and vanilla networks (labelled SGD in experiments). All DUNs are trained using VI (3). The likelihood term in the MFVI ELBO is estimated with 3 MC samples per input. For MFVI and Dropout,

Table 3: Per-dataset HPO configurations.

| DATASET | MIN BUDGET | MAX BUDGET | EARLY STOP PATIENCE | VAL PROP |
|---|---|---|---|---|
| Boston | 200 | 2000 | 200 | 0.15 |
| Concrete | 200 | 2000 | 200 | 0.15 |
| Energy | 200 | 2000 | 200 | 0.15 |
| Kin8nm | 50 | 500 | 50 | 0.15 |
| Naval | 50 | 500 | 50 | 0.15 |
| Power | 50 | 500 | 50 | 0.15 |
| Protein | 50 | 500 | 50 | 0.15 |
| Wine | 100 | 1000 | 100 | 0.15 |
| Yacht | 200 | 2000 | 200 | 0.15 |
| Boston Gap | 200 | 2000 | 200 | 0.15 |
| Concrete Gap | 200 | 2000 | 200 | 0.15 |
| Energy Gap | 200 | 2000 | 200 | 0.15 |
| Kin8nm Gap | 50 | 500 | 50 | 0.15 |
| Naval Gap | 50 | 500 | 50 | 0.15 |
| Power Gap | 50 | 500 | 50 | 0.15 |
| Protein Gap | 50 | 500 | 50 | 0.15 |
| Wine Gap | 100 | 1000 | 100 | 0.15 |
| Yacht Gap | 200 | 2000 | 200 | 0.15 |
| Flights | 2 | 25 | 5 | 0.05 |

10 MC samples are used to estimate the test log-likelihood. Ensembles use 5 elements for prediction. Ensemble elements differ from each other in their initialisation, which is sampled from the He initialisation distribution (He et al., 2015). We do not use adversarial training as, inline with Ashukha et al. (2020), we do not find it to improve results.

Table 4: Hyperparameters optimised for each method.

| HYPERPARAMETER | DUN | SGD | MFVI | MC DROPOUT |
|---|---|---|---|---|
| Learning Rate | ✓ | ✓ | ✓ | ✓ |
| SGD Momentum | ✓ | ✓ | ✓ | ✓ |
| Num. Layers | ✓ | ✓ | ✓ | ✓ |
| Weight Decay | ✓ | ✓ | | ✓ |
| Prior Std. Dev. | | | ✓ | |
| Drop Prob. | | | | ✓ |

Table 5: BOHB hyperparameter optimisation configurations. All hyperparameters were sampled from uniform distributions.

| HYPERPARAMETER | LOWER | UPPER | DEFAULT | LOG | DATA TYPE |
|---|---|---|---|---|---|
| Learning Rate | $1 \times 10^{-4}$ | 1 | 0.01 | True | float |
| SGD Momentum | 0 | 0.99 | 0.5 | False | float |
| Num. Layers | 1 | 40 | 5 | False | int |
| Weight Decay | $1 \times 10^{-6}$ | 0.1 | $5 \times 10^{-4}$ | True | float |
| Prior Std. Dev. | 0.01 | 10 | 1 | True | float |
| Drop Prob. | $5 \times 10^{-3}$ | 0.5 | 0.2 | True | float |

### E.2.2 Evaluation

The best configuration found for each dataset, method and split is used to re-train a model on the entire original training set. For the flights dataset, which does not come with multiple splits, we repeat this five times. We report mean and standard deviation values across all five. Final run training and test sets are normalised using the mean and variance of the original

training set. Note, however, that the results presented in Section 4.3 are unnormalised. The number of epochs used for final training runs is the number of epochs at which the optimal configuration was found with HPO.

Timing experiments for regression models are performed on a 40 core Intel Xeon CPU E5-2650 v3 2.30GHz. We report computation time for a single batch of size 512, which we evaluate across 5 runs. Ensembles, Dropout and MFVI require multiple forward passes per batch. We report the time taken for all passes to be made. For Ensembles, we also include network loading time.

### E.3 Image Experiments

#### E.3.1 Training

The results shown in Section 4.4 are obtained by training ResNet-50 models using SGD with momentum. The initial learning rate, momentum, and weight decay are 0.1, 0.9, and $1 \times 10^{-4}$, respectively. We train on 2 Nvidia P100 GPUs with a batch size of 256 for all experiments. Each dataset is trained for a different number of epochs, shown in Table 6. We decay the learning rate by a factor of 10 at scheduled epochs, also shown in Table 6. Otherwise, all methods and datasets share hyperparameters. These hyperparameter settings are the defaults provided by `PyTorch` for training on ImageNet. We found them to perform well across the board. We report results obtained at the final training epoch. We do not use a separate validation set to determine the best epoch as we found ResNet-50 to not overfit with the chosen schedules.

Table 6: Per-dataset training configuration for image experiments.

| DATASET | No. EPOCHS | LR SCHEDULE |
|---------|------------|-------------|
| MNIST | 90 | 40, 70 |
| Fashion | 90 | 40, 70 |
| SVHN | 90 | 50, 70 |
| CIFAR10 | 300 | 150, 225 |
| CIFAR100 | 300 | 150, 225 |

For dropout experiments, we add dropout to the standard ResNet-50 model (He et al., 2016a) in between the $2^{nd}$ and $3^{rd}$ convolutions in the bottleneck blocks. This approach follows Zagoruyko and Komodakis (2016) and Ashukha et al. (2020) who add dropout in-between the two convolutions of a WideResNet-50's basic block. Following their approach, we try a dropout probability of 0.3. However, we find that this value is too large and causes underfitting. A dropout probability of 0.1 provides stronger results. We show results with both settings in Appendix F.3. We use 10 MC samples for predictions. Ensembles use 5 elements for prediction. Ensemble elements differ from each other in their initialisation, which is sampled from the He initialisation distribution (He et al., 2015). We do not use adversarial training as, inline with Ashukha et al. (2020), we do not find it to improve results.

We modify the standard ResNet-50 architecture such that the first $7 \times 7$ convolution is replaced with a $3 \times 3$ convolution. Additionally, we remove the first max-pooling layer. Following Goyal et al. (2017), we zero-initialise the last batch normalisation layer in residual blocks so that they act as identity functions at the start of training. Because the output block of a ResNet expects to receive activations with a fixed number of channels, we add *adaption layers*. We implement these using $1 \times 1$ convolutions. See Appendix H.2 for an example implementation. Figure 11 shows this modified computational model. Note, however, that this channel mismatch issue is a specific instance of a more general problem of shape and size mismatch between layers in a DUN. Consider the following cases where constructing a DUN requires adaption layers:

- A NN consisting of series of dense layers of different dimensions. E.g. an auto-encoder or U-Net.
- A NN consisting of a mix of convolutional and dense layers. E.g. LeNet (LeCun et al., 1998).

In the first case, we will have dimensionality mismatches between the different dense layers. In the second case, we will have shape mismatches between the 3D convolutional layers and the 1D dense layers, in addition to the potential size mismatches between layers of the same type. Fortunately, adaption layers can be used to solve any shape and size mismatches. Size mismatches can be naively solved by either padding or cropping tensors as appropriate. Another solution is to use (parameter) cheap[3] $1 \times 1$ convolution layers and low-rank[4] dense layers in the case of mismatches between numbers of channels and numbers of dimensions, respectively.

Figure 11: For network architectures in which the input and output number of channels or dimensions is not constant, we add *adaption layers* to the computational model shown in Figure 2. The $n^{\text{th}}$ adaption layer $a_n$ takes a number of channels/dimensions $l_{n-1}$ and outputs $l_n$ channels/dimensions. Later adaption layers are reused multiple times, reducing the number of parameters required. Note that block sizes are unrelated to their number of parameters.

For the MNIST and Fashion-MNIST datasets, we train DUNs with a fixed approximate posterior $q_{\boldsymbol{\alpha}}(d) = p_{\boldsymbol{\beta}}(d)$ for the first 3 epochs. These are the simplest image dataset we work with and can be readily solved with shallower models than ResNet-50. By fixing, $q_{\boldsymbol{\alpha}}(d)$ for the first epochs, we ensure all layers receive strong gradients and become useful for making predictions.

### E.3.2 Evaluation

All methods are trained 5 times on each dataset, allowing for error bars in experiments. We report mean values and standard deviations.

To evaluate the methods' resilience to out of distribution data, we follow Snoek et al. (2019). We train each method on MNIST and evaluate their predictive distributions on increasingly rotated digits. We also train models on CIFAR10 and evaluate them on data submitted to 16 different corruptions (Hendrycks and Dietterich, 2019) with 5 levels of severity each. Per severity results are provided.

We simulate a realistic OOD rejection scenario (Filos et al., 2019) by jointly evaluating our models on an in-distribution and an OOD test set. We allow our methods to reject increasing proportions of the data based on predictive entropy before classifying the rest. All predictions on OOD samples are treated as incorrect. In the main text we provide results with CIFAR10-SVHN as the in-out of distribution dataset pair. Results for the other pairs are found in Appendix F.3. We also perform OOD detection experiments, where we evaluate methods' capacity to distinguish in-distribution and OOD points using predictive entropy.

For all datasets, we compute run times per batch of size of 256 samples on two P100 GPUs. Results are obtained as averages of 5 independent runs. Ensembles and Dropout require multiple forward passes per batch. We report the time taken for all passes to be made. For

Ensembles, we also include network loading time. This is because, in most cases, keeping 5 ResNet-50's in memory is unrealistic.

## E.4 Datasets

We employ the following datasets in Section 4. These are summarized in Table 7.

**Regression:**

- UCI with standard splits (Hernández-Lobato and Adams, 2015)
- UCI with gap splits (Foong et al., 2019b)
- Flights (Airline Delay) (Hensman et al., 2013)

**Image Classification:**

- MNIST (LeCun et al., 1998)
- Fashion-MNIST (Xiao et al., 2017)
- Kuzushiji-MNIST (Clanuwat et al., 2018)
- CIFAR10/100 (Krizhevsky et al., 2009) and Corrupted CIFAR (Hendrycks and Dietterich, 2019)
- SVHN (Netzer et al., 2011)

Table 7: Summary of datasets. For non-UCI datasets, the test and train set sizes are shown in brackets, e.g. (test & train). For the standard UCI splits 90% of the data is used for training and 10% for validation. For the gap splits 66% is used for training and 33% for validation. Note that for the UCI datasets, only the standard number of splits are given since the number of gap splits is equal to the input dimensionality.

| Name | Size | Input Dim. | No. Classes | No. Splits |
|---|---|---|---|---|
| Boston Housing | 506 | 13 | – | 20 |
| Concrete Strength | 1,030 | 8 | – | 20 |
| Energy Efficiency | 768 | 8 | – | 20 |
| Kin8nm | 8,192 | 8 | – | 20 |
| Naval Propulsion | 11,934 | 16 | – | 20 |
| Power Plant | 9,568 | 4 | – | 20 |
| Protein Structure | 45,730 | 9 | – | 5 |
| Wine Quality Red | 1,599 | 11 | – | 20 |
| Yacht Hydrodynamics | 308 | 6 | – | 20 |
| Airline Delay | 2,055,733 (1,955,733 & 100,000) | 8 | – | 2 |
| MNIST | 70,000 (60,000 & 10,000) | 784 ($28 \times 28$) | 10 | 2 |
| Fashion-MNIST | 70,000 (60,000 & 10,000) | 784 ($28 \times 28$) | 10 | 2 |
| Kuzushiji-MNIST | 70,000 (60,000 & 10,000) | 784 ($28 \times 28$) | 10 | 2 |
| CIFAR10 | 60,000 (50,000 & 10,000) | 3072 ($32 \times 32 \times 3$) | 10 | 2 |
| CIFAR100 | 60,000 (50,000 & 10,000) | 3072 ($32 \times 32 \times 3$) | 100 | 2 |
| SVHN | 99,289 (73,257 & 26,032) | 3072 ($32 \times 32 \times 3$) | 10 | 2 |

# F   Additional Results

## F.1 Toy Datasets

In addition to the 1D toy dataset from Izmailov et al. (2019) and the Wiggle dataset introduced in Section 4.2, we conduct experiments on another three 1D toy datasets. Similarly to that of Izmailov et al. (2019), the first of these datasets is composed of three disjoint clusters of inputs. However, these are arranged such that they can be fit by slower varying functions. We dub it "Simple_1d". The second is the toy dataset used by Foong et al. (2019b) to evaluate the capacity of NN approximate inference techniques to express model uncertainty in between disjoint clusters of data, also know as "in-between" uncertainty. The third is generated by sampling a function from a GP with a Matern kernel with additive Gaussian

noise. We dub it "Matern". We show all 5 1D toy datasets in Figure 12, where we fit them with a GP.

Figure 12: Fit obtained by a GP with an RBF covariance function on the following datasets, from left to right: Simple_1d, (Izmailov et al., 2019), (Foong et al., 2019b), Matern, Wiggle. Error bars represent the standard deviations of the distributions over functions.

### F.1.1 Different Depths

In this section, we evaluate the effects of network depth on uncertainty estimates. We first train DUNs of depths 5, 10 and 15 on all 1d toy datasets. The results are shown in Figure 13. DUNs are able to fit all of the datasets well. However, the 5 layer versions provide noticeably smaller uncertainty estimates in between clusters of data. The uncertainty estimates from DUNs benefit from depth for 2 reasons: increased depth means increasing the number of explanations of the data over which we perform BMA and deeper subnetworks are able to express faster varying functions, which are more likely to disagree with each other.

Figure 13: Increasing depth DUNs trained on all 1d toy datasets. Each row corresponds to a different dataset. From top to bottom: Simple_1d, (Izmailov et al., 2019), (Foong et al., 2019b), Matern, Wiggle.

We also train each of our NN-based baselines with depths 1, 2 and 3 on each of these datasets. Recall that by depth, we refer to the number of hidden layers. Results are shown in Figure 14.

Foong et al. (2019b) prove that single hidden layer MFVI and dropout networks are unable to express high uncertainty in between regions of low uncertainty. Indeed, we observe this in our

results. Further inline with the author's empirical observations, we find that deeper networks also fail to represent uncertainty in between clusters of points when making use of these approximate inference methods. Interestingly, the size of the error bars in the extrapolation regime seems to grow with depth for MFVI but shrink when using dropout. The amount of in-between and extrapolation uncertainty expressed by deep ensembles grows with depth. We attribute this to deeper models being able to express a wider range of functions, thus creating more opportunities for disagreement.

Shallower dropout models tend to underfit faster varying functions, like Matern and Wiggle. For the latter, even the 3 hidden layer model underfits slightly, failing to capture the effects of the faster varying, lower amplitude sinusoid. MFVI completely fails to fit fast varying functions, even for deeper networks. Additionally, the functions it learns look piecewise linear. This might be the result of variational overpruning (Trippe and Turner, 2018). Ensembles are able to fit all datasets well.

Figure 14: NN baselines fit on all toy datasets.

### F.1.2 Overcounting Data with MFVI

In an attempt to fit MFVI networks to faster varying functions, we overcount the importance of the data in the ELBO. This type of objective targets what is often referred to as a tempered posterior (Wenzel et al., 2020a): $p_{\text{overcount}}(\mathbf{w}|\mathfrak{D}) \propto p(\mathfrak{D}|\mathbf{w})^T p(\mathbf{w})$.

$$\text{ELBO}_{\text{overcount}} = -KL(q(\mathbf{w})||p(\mathbf{w})) + T \cdot \mathbb{E}_{q(\mathbf{w})}[\sum_{n=1}^{N} p(y^{(n)}|\mathbf{x}^{(n)}, \mathbf{w})]$$

We experiment by setting the overcounting factor $T$ to the values: 1, 4 and 16. The results are shown in Figure 15. Although increasing the relative importance of the data dependent likelihood term in the ELBO helps MFVI fit the Matern dataset and the dataset from Foong et al. (2019b), the method still fails to fit Wiggle. Overcounting the data results in smaller error bars.

Figure 15: MFVI networks fit on all toy datasets for different overcount settings.

### F.1.3 2d Toy Datasets

We evaluate the approaches under consideration on two 2d toy datasets, taken from Foong et al. (2019a). These are dubbed Axis, Figure 16, and Origin, Figure 17. We employ 15 hidden layers with DUNs and 3 hidden layers with all other approaches.

DUNs and ensembles do not provide significantly increased uncertainty estimates in the regions between clusters of data on the Axis dataset. Both methods perform well on Origin. Otherwise, all methods display similar properties as in previous sections.

Figure 16: All methods under consideration trained on the Axis dataset. The top row shows the standard deviation values provided by each method for each point in the 2d input space. The bottom plot shows each method's predictions on a cross section of the input space at $x_2=0$. From left to right, the following methods are shown: dropout, MFVI, DUN, deep ensembles, GP.

Figure 17: All methods under consideration trained on the Origin dataset. The top row shows the standard deviation values provided by each method for each point in the 2d input space. The bottom plot shows each method's predictions on a cross section of the input space. From left to right, the following methods are shown: dropout, MFVI, DUN, deep ensembles, GP.

### F.1.4 Non-residual Models

We employ residual architectures for most experiments in this work. This subsection explores the effect of residual connections on DUNs. We first fit non-residual (MLP) DUNs on all of our 1d toy datasets. The results are given in Figure 18. The learnt functions resemble those obtained with residual networks in Figure 13. However, non-residual DUNs tend to provide less consistent uncertainty estimates in the extrapolation regime, especially when working with shallower models.

We further compare the in-distribution fits from residual DUNs, MLP DUNs, and deep ensembles in Figure 19. Ensemble elements differ slightly from each other in their predictions within the data dense regions. These predictions are averaged, making for mostly smooth functions. Functions expressed at most depths of the MLP DUNs seem to vary together rapidly within the data region. Their mean prediction also varies rapidly, suggesting overfitting. In an MLP architecture, each successive layer only receives the previous one's output as its input. We hypothesize that, because of this structure, once a layer overfits a

Figure 18: DUNs with an MLP architecture trained on 1d toy datasets.

data point, the following layer is unlikely to modify the function in the area of that data point, as that would increase the training loss. This leads to most subnetworks only disagreeing about their predictions out of distribution. Functions expressed by residual DUNs differ somewhat in-distribution, allowing some robustness to overfitting. We hypothesize that this occurs because each layer takes a linear combination of all previous layers' activations as its input. This prevents re-using the previous subnetworks' fits.

Ensembles provide diverse explanations both in and out of distribution. This results in both better accuracy and predictive uncertainty than single models. DUNs provide explanations which differ from each other mostly out of distribution. They provide uncertainty estimates out of distribution but their accuracy on in-distribution points is similar to that of a single model.

Figure 19: We fit the Simple_1d toy dataset with 15 a layer MLP DUN, a 15 layer residual DUN and a 20 network deep ensemble with 3 hidden layers per network. The leftmost plot shows mean predictions and standard deviations corresponding to model uncertainty. The rightmost plot shows individual predictions from DUN subnetworks and ensemble elements.

## F.2 Regression

In Section 4.3, we discussed the performance of DUNs compared with SGD, Dropout, Ensembles, and MFVI, in terms of LL, RMSE, TCE, and batch time. In this section, we elaborate by providing an additional metric: Regression Calibration Error (RCE), discussed in Appendix D. We also further investigate the predictive performance to prediction time trade-off and provide results for the UCI gap splits.

UCI standard split results are found in Figure 20. As before, we rank methods from 1 to 5 based on mean performance, reporting mean ranks and standard deviations. Dropout obtains the best mean rank in terms of RMSE, followed closely by Ensembles. DUNs are third, significantly ahead of MFVI and SGD. Even so, DUNs outperform Dropout and Ensembles in terms of TCE, i.e. DUNs more reliably assign large error bars to points on which they make incorrect predictions. Consequently, in terms of LL, a metric which considers both uncertainty and accuracy, DUNs perform competitively (the LL rank distributions for all three methods overlap almost completely). However, on an alternate uncertainty metric, RCE, Dropout tends to outperform DUNs. This is indicative that the Dropout predictive posterior is better approximated by a Gaussian than DUNs' predictive posterior. Ensembles still performs poorly and is only better than SGD. MFVI provides the best calibrated uncertainty estimates according to TCE and ties with Dropout according to RCE. Despite this, its mean predictions are inaccurate, as evidenced by it being last in terms of RMSE. This leads to MFVI's LL rank only being better than SGD's.

Figure 21 shows results for gap splits, designed to evaluate methods' capacity to express in-between uncertainty. All methods tend to perform worse in terms of the predictive performance metrics, indicating that the gap splits represent a more challenging problem. This trend is exemplified in the naval, and to a lesser extent, the energy datasets. Here, DUNs outperform Dropout in terms of LL rank. However, they are both outperformed by MFVI and Ensembles. DUNs consistently outperform multiple forward pass methods in terms of prediction time.

In Figure 22, we show LL vs batch time Pareto curves for all methods under consideration on the UCI datasets with standard splits. DUNs are Pareto efficient in 5 datasets, performing competitively in all of them. Dropout and Ensembles also tend to perform well.

Finally, in Table 8 and Table 9, we provide mean and standard deviation results for both UCI standard and gap splits.

Figure 20: Quartiles for results on UCI regression datasets across *standard splits*. Average ranks are computed across datasets. For LL, higher is better. Otherwise, lower is better.

Figure 21: Quartiles for results on UCI regression datasets across *gap splits*. Average ranks are computed across datasets. For LL, higher is better. Otherwise, lower is better.

Figure 22: Pareto frontiers showing mean LL vs batch prediction time on the UCI datasets with standard splits. MFVI and Dropout are shown for 5, 10, 20, and 30 samples. Ensembles are shown with 1, 2, 3, 4, and 5 elements. Note that a single element ensemble is equivalent to SGD. Top left is better. Bottom right is worse. Timing includes overhead such as ensemble element loading.

Table 8: Mean values and standard deviations for results on UCI regression datasets across *standard splits*. Bold blue text denotes the best mean value for each dataset and each metric. Bold red text denotes the worst mean value. Note that in some cases the best/worst mean values are within error of other mean values.

| METRIC | METHOD DATASET | DUN | DUN (MLP) | DROPOUT | ENSEMBLE | MFVI | SGD |
|---|---|---|---|---|---|---|---|
| LL ↑ | boston | $-2.604_{\pm0.351}$ | $-2.604_{\pm0.368}$ | $-2.882_{\pm1.028}$ | $\mathbf{-2.454}_{\pm0.275}$ | $-2.573_{\pm0.136}$ | $\mathbf{-2.942}_{\pm0.676}$ |
| | concrete | $-3.005_{\pm0.212}$ | $-3.051_{\pm0.278}$ | $-3.051_{\pm0.308}$ | $\mathbf{-2.886}_{\pm0.153}$ | $-3.190_{\pm0.110}$ | $\mathbf{-3.214}_{\pm0.399}$ |
| | energy | $-1.037_{\pm0.159}$ | $-1.564_{\pm0.383}$ | $\mathbf{-0.975}_{\pm0.509}$ | $-1.298_{\pm0.210}$ | $\mathbf{-1.961}_{\pm0.648}$ | $-1.348_{\pm0.225}$ |
| | kin8nm | $1.151_{\pm0.083}$ | $1.111_{\pm0.103}$ | $\mathbf{1.231}_{\pm0.086}$ | $0.813_{\pm1.224}$ | $1.055_{\pm0.084}$ | $\mathbf{0.905}_{\pm0.778}$ |
| | naval | $4.245_{\pm1.108}$ | $4.472_{\pm1.239}$ | $\mathbf{5.429}_{\pm0.735}$ | $5.081_{\pm0.156}$ | $\mathbf{3.389}_{\pm2.891}$ | $4.821_{\pm0.621}$ |
| | power | $-2.695_{\pm0.086}$ | $-2.719_{\pm0.069}$ | $-2.790_{\pm0.118}$ | $\mathbf{-2.663}_{\pm0.055}$ | $\mathbf{-2.877}_{\pm0.041}$ | $-2.733_{\pm0.081}$ |
| | protein | $-2.657_{\pm0.044}$ | $-2.692_{\pm0.020}$ | $-2.623_{\pm0.036}$ | $\mathbf{-2.561}_{\pm0.026}$ | $\mathbf{-2.929}_{\pm0.038}$ | $-2.717_{\pm0.064}$ |
| | wine | $-1.031_{\pm0.119}$ | $\mathbf{-0.979}_{\pm0.113}$ | $-1.003_{\pm0.128}$ | $-1.116_{\pm0.582}$ | $-1.007_{\pm0.063}$ | $\mathbf{-1.212}_{\pm0.485}$ |
| | yacht | $-2.420_{\pm0.523}$ | $-2.463_{\pm0.197}$ | $\mathbf{-1.330}_{\pm0.436}$ | $-2.441_{\pm0.189}$ | $-2.238_{\pm0.952}$ | $\mathbf{-2.525}_{\pm0.354}$ |
| RMSE ↓ | boston | $3.200_{\pm0.978}$ | $3.157_{\pm0.885}$ | $\mathbf{2.832}_{\pm0.768}$ | $2.835_{\pm0.808}$ | $\mathbf{3.218}_{\pm0.837}$ | $\mathbf{3.218}_{\pm0.904}$ |
| | concrete | $4.613_{\pm0.607}$ | $4.571_{\pm0.703}$ | $4.610_{\pm0.572}$ | $\mathbf{4.552}_{\pm0.582}$ | $\mathbf{5.894}_{\pm0.742}$ | $4.983_{\pm0.914}$ |
| | energy | $0.612_{\pm0.157}$ | $0.948_{\pm0.474}$ | $0.571_{\pm0.204}$ | $\mathbf{0.507}_{\pm0.110}$ | $\mathbf{1.686}_{\pm1.016}$ | $0.797_{\pm0.283}$ |
| | kin8nm | $0.076_{\pm0.005}$ | $0.077_{\pm0.006}$ | $\mathbf{0.070}_{\pm0.005}$ | $\mathbf{0.304}_{\pm0.991}$ | $0.084_{\pm0.007}$ | $0.202_{\pm0.544}$ |
| | naval | $0.003_{\pm0.002}$ | $0.002_{\pm0.001}$ | $\mathbf{0.001}_{\pm0.001}$ | $\mathbf{0.001}_{\pm0.000}$ | $\mathbf{0.005}_{\pm0.005}$ | $0.002_{\pm0.001}$ |
| | power | $3.573_{\pm0.254}$ | $3.671_{\pm0.247}$ | $3.823_{\pm0.350}$ | $\mathbf{3.444}_{\pm0.238}$ | $\mathbf{4.286}_{\pm0.179}$ | $3.697_{\pm0.272}$ |
| | protein | $3.402_{\pm0.058}$ | $3.412_{\pm0.076}$ | $3.425_{\pm0.070}$ | $\mathbf{3.260}_{\pm0.074}$ | $\mathbf{4.511}_{\pm0.145}$ | $3.589_{\pm0.174}$ |
| | wine | $0.659_{\pm0.061}$ | $0.629_{\pm0.047}$ | $\mathbf{0.642}_{\pm0.049}$ | $\mathbf{1.934}_{\pm5.708}$ | $0.660_{\pm0.040}$ | $0.652_{\pm0.054}$ |
| | yacht | $2.514_{\pm1.985}$ | $2.465_{\pm0.841}$ | $\mathbf{0.876}_{\pm0.411}$ | $1.429_{\pm0.483}$ | $\mathbf{3.419}_{\pm7.333}$ | $2.352_{\pm0.905}$ |
| RCE ↓ | boston | $0.045_{\pm0.016}$ | $\mathbf{0.043}_{\pm0.013}$ | $\mathbf{0.058}_{\pm0.037}$ | $0.046_{\pm0.015}$ | $0.049_{\pm0.014}$ | $0.052_{\pm0.032}$ |
| | concrete | $0.037_{\pm0.011}$ | $0.039_{\pm0.011}$ | $0.036_{\pm0.011}$ | $\mathbf{0.053}_{\pm0.020}$ | $\mathbf{0.030}_{\pm0.008}$ | $0.040_{\pm0.016}$ |
| | energy | $0.064_{\pm0.031}$ | $0.120_{\pm0.069}$ | $\mathbf{0.059}_{\pm0.047}$ | $\mathbf{0.157}_{\pm0.052}$ | $0.070_{\pm0.051}$ | $0.072_{\pm0.031}$ |
| | kin8nm | $\mathbf{0.014}_{\pm0.007}$ | $0.021_{\pm0.014}$ | $\mathbf{0.014}_{\pm0.006}$ | $\mathbf{0.090}_{\pm0.199}$ | $0.016_{\pm0.007}$ | $0.028_{\pm0.051}$ |
| | naval | $0.134_{\pm0.102}$ | $0.094_{\pm0.123}$ | $0.100_{\pm0.074}$ | $\mathbf{0.191}_{\pm0.079}$ | $0.087_{\pm0.108}$ | $\mathbf{0.072}_{\pm0.073}$ |
| | power | $0.016_{\pm0.004}$ | $0.018_{\pm0.005}$ | $0.015_{\pm0.012}$ | $\mathbf{0.032}_{\pm0.005}$ | $\mathbf{0.010}_{\pm0.003}$ | $0.017_{\pm0.005}$ |
| | protein | $0.048_{\pm0.005}$ | $0.045_{\pm0.003}$ | $0.043_{\pm0.006}$ | $\mathbf{0.055}_{\pm0.007}$ | $\mathbf{0.014}_{\pm0.003}$ | $0.041_{\pm0.011}$ |
| | wine | $0.030_{\pm0.009}$ | $0.031_{\pm0.013}$ | $\mathbf{0.027}_{\pm0.013}$ | $\mathbf{0.100}_{\pm0.214}$ | $0.028_{\pm0.009}$ | $0.083_{\pm0.195}$ |
| | yacht | $0.141_{\pm0.078}$ | $0.177_{\pm0.066}$ | $\mathbf{0.117}_{\pm0.068}$ | $\mathbf{0.311}_{\pm0.089}$ | $0.156_{\pm0.190}$ | $0.153_{\pm0.085}$ |
| TCE ↓ | boston | $0.053_{\pm0.034}$ | $\mathbf{0.047}_{\pm0.030}$ | $\mathbf{0.089}_{\pm0.076}$ | $0.055_{\pm0.038}$ | $0.060_{\pm0.035}$ | $0.082_{\pm0.075}$ |
| | concrete | $0.054_{\pm0.027}$ | $0.048_{\pm0.025}$ | $0.047_{\pm0.028}$ | $\mathbf{0.067}_{\pm0.032}$ | $\mathbf{0.036}_{\pm0.020}$ | $0.060_{\pm0.045}$ |
| | energy | $\mathbf{0.072}_{\pm0.073}$ | $0.103_{\pm0.112}$ | $0.088_{\pm0.085}$ | $\mathbf{0.221}_{\pm0.101}$ | $0.097_{\pm0.090}$ | $0.083_{\pm0.057}$ |
| | kin8nm | $\mathbf{0.024}_{\pm0.022}$ | $0.042_{\pm0.038}$ | $0.025_{\pm0.021}$ | $\mathbf{0.065}_{\pm0.054}$ | $\mathbf{0.024}_{\pm0.015}$ | $0.031_{\pm0.022}$ |
| | naval | $\mathbf{0.212}_{\pm0.159}$ | $0.127_{\pm0.162}$ | $0.153_{\pm0.128}$ | $\mathbf{0.212}_{\pm0.147}$ | $0.118_{\pm0.143}$ | $\mathbf{0.112}_{\pm0.118}$ |
| | power | $0.020_{\pm0.007}$ | $0.022_{\pm0.011}$ | $0.024_{\pm0.026}$ | $\mathbf{0.045}_{\pm0.010}$ | $\mathbf{0.015}_{\pm0.006}$ | $0.020_{\pm0.009}$ |
| | protein | $0.069_{\pm0.012}$ | $0.058_{\pm0.011}$ | $0.063_{\pm0.011}$ | $\mathbf{0.094}_{\pm0.014}$ | $\mathbf{0.020}_{\pm0.008}$ | $0.061_{\pm0.017}$ |
| | wine | $0.051_{\pm0.028}$ | $0.047_{\pm0.033}$ | $0.040_{\pm0.028}$ | $0.088_{\pm0.201}$ | $\mathbf{0.027}_{\pm0.013}$ | $\mathbf{0.109}_{\pm0.197}$ |
| | yacht | $\mathbf{0.122}_{\pm0.119}$ | $0.169_{\pm0.131}$ | $0.131_{\pm0.114}$ | $\mathbf{0.341}_{\pm0.176}$ | $0.196_{\pm0.207}$ | $0.175_{\pm0.130}$ |
| batch time ↓ | boston | $0.003_{\pm0.003}$ | $\mathbf{0.001}_{\pm0.000}$ | $0.018_{\pm0.006}$ | $0.016_{\pm0.004}$ | $\mathbf{0.029}_{\pm0.021}$ | $\mathbf{0.001}_{\pm0.000}$ |
| | concrete | $0.005_{\pm0.003}$ | $\mathbf{0.002}_{\pm0.001}$ | $0.019_{\pm0.007}$ | $0.050_{\pm0.035}$ | $\mathbf{0.055}_{\pm0.042}$ | $0.003_{\pm0.002}$ |
| | energy | $0.007_{\pm0.008}$ | $0.005_{\pm0.002}$ | $0.017_{\pm0.007}$ | $0.037_{\pm0.020}$ | $\mathbf{0.043}_{\pm0.037}$ | $\mathbf{0.002}_{\pm0.001}$ |
| | kin8nm | $0.019_{\pm0.014}$ | $0.011_{\pm0.008}$ | $0.029_{\pm0.009}$ | $0.026_{\pm0.008}$ | $\mathbf{0.157}_{\pm0.097}$ | $\mathbf{0.002}_{\pm0.001}$ |
| | naval | $0.019_{\pm0.010}$ | $0.012_{\pm0.005}$ | $0.029_{\pm0.009}$ | $0.065_{\pm0.032}$ | $\mathbf{0.156}_{\pm0.128}$ | $\mathbf{0.005}_{\pm0.003}$ |
| | power | $0.024_{\pm0.007}$ | $0.016_{\pm0.006}$ | $0.023_{\pm0.006}$ | $0.074_{\pm0.038}$ | $\mathbf{0.138}_{\pm0.106}$ | $\mathbf{0.007}_{\pm0.005}$ |
| | protein | $0.022_{\pm0.008}$ | $0.018_{\pm0.002}$ | $0.024_{\pm0.004}$ | $0.051_{\pm0.022}$ | $\mathbf{0.178}_{\pm0.099}$ | $\mathbf{0.004}_{\pm0.002}$ |
| | wine | $0.046_{\pm0.026}$ | $0.028_{\pm0.009}$ | $0.031_{\pm0.006}$ | $0.046_{\pm0.034}$ | $\mathbf{0.078}_{\pm0.048}$ | $\mathbf{0.004}_{\pm0.003}$ |
| | yacht | $0.004_{\pm0.003}$ | $0.003_{\pm0.002}$ | $0.017_{\pm0.005}$ | $0.035_{\pm0.035}$ | $\mathbf{0.038}_{\pm0.022}$ | $\mathbf{0.002}_{\pm0.002}$ |

Table 9: Mean values and standard deviations for results on UCI regression datasets across *gap splits*. Bold blue text denotes the best mean value for each dataset and each metric. Bold red text denotes the worst mean value. Note that in some cases the best/worst mean values are within error of other mean values.

| METRIC | METHOD DATASET | DUN | DUN (MLP) | DROPOUT | ENSEMBLE | MFVI | SGD |
|---|---|---|---|---|---|---|---|
| LL ↑ | boston | $-3.107_{\pm0.593}$ | $-3.033_{\pm0.409}$ | $-4.001_{\pm1.814}$ | $-3.106_{\pm1.481}$ | $\mathbf{-2.703}_{\pm0.072}$ | $\mathbf{-4.217}_{\pm1.876}$ |
| | concrete | $-4.222_{\pm0.818}$ | $-4.152_{\pm0.433}$ | $\mathbf{-5.170}_{\pm1.376}$ | $-3.631_{\pm0.523}$ | $\mathbf{-3.460}_{\pm0.177}$ | $-4.839_{\pm1.585}$ |
| | energy | $-10.730_{\pm13.477}$ | $-6.477_{\pm7.516}$ | $-8.102_{\pm13.796}$ | $\mathbf{-5.423}_{\pm7.290}$ | $-9.093_{\pm10.573}$ | $\mathbf{-15.295}_{\pm26.058}$ |
| | kin8nm | $1.029_{\pm0.133}$ | $1.110_{\pm0.073}$ | $\mathbf{1.215}_{\pm0.049}$ | $\mathbf{0.315}_{\pm2.397}$ | $0.942_{\pm0.240}$ | $0.991_{\pm0.131}$ |
| | naval | $-16.279_{\pm19.437}$ | $-19.165_{\pm11.324}$ | $\mathbf{-523.856}_{\pm570.116}$ | $\mathbf{-4.573}_{\pm7.496}$ | $-15.208_{\pm43.758}$ | $-60.470_{\pm53.213}$ |
| | power | $-2.998_{\pm0.325}$ | $-2.961_{\pm0.089}$ | $\mathbf{-3.178}_{\pm0.224}$ | $\mathbf{-2.904}_{\pm0.110}$ | $-2.980_{\pm0.184}$ | $-3.022_{\pm0.141}$ |
| | protein | $\mathbf{-3.835}_{\pm0.998}$ | $-3.553_{\pm0.371}$ | $-3.459_{\pm0.444}$ | $\mathbf{-3.071}_{\pm0.261}$ | $-3.083_{\pm0.086}$ | $-3.554_{\pm0.408}$ |
| | wine | $-1.417_{\pm0.474}$ | $\mathbf{-2.121}_{\pm1.227}$ | $-1.267_{\pm0.677}$ | $-1.126_{\pm0.137}$ | $\mathbf{-0.965}_{\pm0.033}$ | $-2.026_{\pm1.130}$ |
| | yacht | $-2.122_{\pm0.584}$ | $-2.165_{\pm0.235}$ | $-2.344_{\pm0.995}$ | $\mathbf{-2.568}_{\pm1.250}$ | $\mathbf{-2.114}_{\pm0.399}$ | $-2.442_{\pm0.520}$ |
| RMSE ↓ | boston | $3.636_{\pm0.493}$ | $3.585_{\pm0.517}$ | $3.597_{\pm0.684}$ | $\mathbf{3.512}_{\pm0.573}$ | $3.756_{\pm0.418}$ | $\mathbf{4.593}_{\pm2.927}$ |
| | concrete | $7.196_{\pm0.821}$ | $7.461_{\pm0.948}$ | $7.064_{\pm0.921}$ | $\mathbf{6.853}_{\pm0.796}$ | $\mathbf{7.548}_{\pm0.865}$ | $7.367_{\pm0.866}$ |
| | energy | $2.938_{\pm3.017}$ | $3.606_{\pm3.927}$ | $\mathbf{2.874}_{\pm2.254}$ | $3.364_{\pm3.696}$ | $\mathbf{8.614}_{\pm9.390}$ | $3.061_{\pm2.880}$ |
| | kin8nm | $0.080_{\pm0.006}$ | $0.078_{\pm0.005}$ | $\mathbf{0.071}_{\pm0.003}$ | $\mathbf{1.632}_{\pm4.418}$ | $0.095_{\pm0.025}$ | $0.085_{\pm0.007}$ |
| | naval | $0.022_{\pm0.014}$ | $0.021_{\pm0.007}$ | $\mathbf{0.034}_{\pm0.018}$ | $\mathbf{0.018}_{\pm0.009}$ | $0.033_{\pm0.041}$ | $0.020_{\pm0.009}$ |
| | power | $\mathbf{4.299}_{\pm0.416}$ | $4.584_{\pm0.356}$ | $\mathbf{4.688}_{\pm0.335}$ | $4.369_{\pm0.383}$ | $4.680_{\pm0.703}$ | $4.621_{\pm0.339}$ |
| | protein | $\mathbf{5.206}_{\pm0.780}$ | $5.101_{\pm0.526}$ | $5.133_{\pm0.636}$ | $\mathbf{4.801}_{\pm0.599}$ | $5.115_{\pm0.298}$ | $5.171_{\pm0.632}$ |
| | wine | $0.697_{\pm0.043}$ | $0.692_{\pm0.041}$ | $0.660_{\pm0.040}$ | $0.673_{\pm0.039}$ | $\mathbf{0.632}_{\pm0.029}$ | $\mathbf{0.731}_{\pm0.070}$ |
| | yacht | $1.851_{\pm0.750}$ | $1.852_{\pm0.623}$ | $\mathbf{2.290}_{\pm2.108}$ | $\mathbf{1.841}_{\pm0.836}$ | $1.836_{\pm0.712}$ | $2.214_{\pm0.793}$ |
| RCE ↓ | boston | $0.072_{\pm0.047}$ | $0.068_{\pm0.038}$ | $0.126_{\pm0.103}$ | $0.103_{\pm0.240}$ | $\mathbf{0.031}_{\pm0.014}$ | $\mathbf{0.156}_{\pm0.236}$ |
| | concrete | $0.108_{\pm0.066}$ | $0.097_{\pm0.034}$ | $\mathbf{0.155}_{\pm0.072}$ | $0.057_{\pm0.043}$ | $\mathbf{0.030}_{\pm0.016}$ | $0.134_{\pm0.075}$ |
| | energy | $0.120_{\pm0.149}$ | $\mathbf{0.095}_{\pm0.067}$ | $0.117_{\pm0.094}$ | $0.128_{\pm0.076}$ | $\mathbf{0.187}_{\pm0.158}$ | $0.162_{\pm0.180}$ |
| | kin8nm | $0.027_{\pm0.021}$ | $0.015_{\pm0.012}$ | $\mathbf{0.012}_{\pm0.010}$ | $\mathbf{0.137}_{\pm0.308}$ | $0.017_{\pm0.011}$ | $0.024_{\pm0.023}$ |
| | naval | $0.580_{\pm0.321}$ | $0.649_{\pm0.293}$ | $0.719_{\pm0.314}$ | $\mathbf{0.499}_{\pm0.347}$ | $0.525_{\pm0.353}$ | $\mathbf{0.732}_{\pm0.278}$ |
| | power | $0.027_{\pm0.039}$ | $0.013_{\pm0.006}$ | $\mathbf{0.046}_{\pm0.026}$ | $\mathbf{0.010}_{\pm0.005}$ | $0.019_{\pm0.017}$ | $0.023_{\pm0.017}$ |
| | protein | $0.087_{\pm0.053}$ | $0.076_{\pm0.027}$ | $0.062_{\pm0.030}$ | $\mathbf{0.034}_{\pm0.021}$ | $\mathbf{0.217}_{\pm0.387}$ | $0.079_{\pm0.034}$ |
| | wine | $0.076_{\pm0.057}$ | $\mathbf{0.134}_{\pm0.096}$ | $0.047_{\pm0.068}$ | $0.033_{\pm0.017}$ | $\mathbf{0.023}_{\pm0.009}$ | $0.114_{\pm0.094}$ |
| | yacht | $0.085_{\pm0.036}$ | $\mathbf{0.075}_{\pm0.024}$ | $0.137_{\pm0.173}$ | $\mathbf{0.249}_{\pm0.312}$ | $0.102_{\pm0.052}$ | $0.077_{\pm0.045}$ |
| TCE ↓ | boston | $0.138_{\pm0.092}$ | $0.132_{\pm0.065}$ | $0.221_{\pm0.154}$ | $0.134_{\pm0.235}$ | $\mathbf{0.037}_{\pm0.023}$ | $\mathbf{0.234}_{\pm0.231}$ |
| | concrete | $0.212_{\pm0.094}$ | $0.199_{\pm0.055}$ | $\mathbf{0.280}_{\pm0.098}$ | $0.107_{\pm0.088}$ | $\mathbf{0.052}_{\pm0.040}$ | $0.248_{\pm0.107}$ |
| | energy | $0.175_{\pm0.211}$ | $\mathbf{0.161}_{\pm0.132}$ | $0.180_{\pm0.149}$ | $0.216_{\pm0.137}$ | $\mathbf{0.267}_{\pm0.208}$ | $0.227_{\pm0.238}$ |
| | kin8nm | $0.064_{\pm0.051}$ | $0.030_{\pm0.031}$ | $\mathbf{0.025}_{\pm0.030}$ | $\mathbf{0.150}_{\pm0.303}$ | $0.029_{\pm0.027}$ | $0.048_{\pm0.051}$ |
| | naval | $0.650_{\pm0.284}$ | $0.714_{\pm0.229}$ | $0.744_{\pm0.296}$ | $\mathbf{0.560}_{\pm0.334}$ | $0.585_{\pm0.336}$ | $\mathbf{0.763}_{\pm0.253}$ |
| | power | $0.055_{\pm0.088}$ | $0.033_{\pm0.017}$ | $\mathbf{0.109}_{\pm0.049}$ | $\mathbf{0.020}_{\pm0.015}$ | $0.034_{\pm0.039}$ | $0.057_{\pm0.040}$ |
| | protein | $0.167_{\pm0.088}$ | $0.157_{\pm0.052}$ | $0.129_{\pm0.057}$ | $\mathbf{0.071}_{\pm0.050}$ | $\mathbf{0.240}_{\pm0.375}$ | $0.159_{\pm0.064}$ |
| | wine | $0.153_{\pm0.104}$ | $\mathbf{0.233}_{\pm0.135}$ | $0.084_{\pm0.115}$ | $0.073_{\pm0.046}$ | $\mathbf{0.019}_{\pm0.007}$ | $0.209_{\pm0.146}$ |
| | yacht | $0.133_{\pm0.058}$ | $\mathbf{0.095}_{\pm0.073}$ | $0.097_{\pm0.066}$ | $\mathbf{0.289}_{\pm0.322}$ | $0.136_{\pm0.107}$ | $0.098_{\pm0.056}$ |
| batch time ↓ | boston | $0.029_{\pm0.018}$ | $0.037_{\pm0.020}$ | $0.032_{\pm0.005}$ | $0.044_{\pm0.016}$ | $\mathbf{0.047}_{\pm0.030}$ | $\mathbf{0.003}_{\pm0.001}$ |
| | concrete | $0.033_{\pm0.018}$ | $0.016_{\pm0.010}$ | $0.022_{\pm0.006}$ | $0.043_{\pm0.030}$ | $\mathbf{0.090}_{\pm0.078}$ | $\mathbf{0.003}_{\pm0.003}$ |
| | energy | $0.021_{\pm0.021}$ | $0.023_{\pm0.014}$ | $0.029_{\pm0.005}$ | $0.025_{\pm0.006}$ | $\mathbf{0.117}_{\pm0.099}$ | $\mathbf{0.002}_{\pm0.000}$ |
| | kin8nm | $0.012_{\pm0.007}$ | $0.008_{\pm0.004}$ | $0.019_{\pm0.007}$ | $0.040_{\pm0.034}$ | $\mathbf{0.244}_{\pm0.157}$ | $\mathbf{0.003}_{\pm0.002}$ |
| | naval | $0.013_{\pm0.006}$ | $0.010_{\pm0.005}$ | $0.024_{\pm0.011}$ | $0.040_{\pm0.018}$ | $\mathbf{0.203}_{\pm0.196}$ | $\mathbf{0.003}_{\pm0.002}$ |
| | power | $0.013_{\pm0.002}$ | $0.018_{\pm0.005}$ | $0.023_{\pm0.007}$ | $0.073_{\pm0.037}$ | $\mathbf{0.215}_{\pm0.125}$ | $\mathbf{0.005}_{\pm0.003}$ |
| | protein | $0.023_{\pm0.006}$ | $0.016_{\pm0.005}$ | $0.027_{\pm0.007}$ | $0.057_{\pm0.030}$ | $\mathbf{0.258}_{\pm0.135}$ | $\mathbf{0.005}_{\pm0.003}$ |
| | wine | $0.013_{\pm0.007}$ | $0.008_{\pm0.002}$ | $0.018_{\pm0.008}$ | $0.065_{\pm0.051}$ | $\mathbf{0.102}_{\pm0.094}$ | $\mathbf{0.005}_{\pm0.004}$ |
| | yacht | $0.003_{\pm0.002}$ | $0.004_{\pm0.002}$ | $0.015_{\pm0.003}$ | $0.015_{\pm0.004}$ | $\mathbf{0.021}_{\pm0.010}$ | $\mathbf{0.001}_{\pm0.000}$ |

## F.3 Image Classification

Table 10 shows a detailed breakdown of the performance of DUNs, as well as various benchmark methods, on image datasets.

Table 10: Mean values and standard deviations for results on image datasets. Bold blue text denotes the best mean value for each dataset and each metric. Bold red text denotes the worst mean value. Note that in some cases the best/worst mean values are within error of other mean values.

| DATASET | METHOD | LL ↑ | ERROR ↓ | BRIER ↓ | ECE ↓ |
|---|---|---|---|---|---|
| CIFAR10 | DUN | $\mathbf{-0.240}_{\pm0.011}$ | $0.056_{\pm0.002}$ | $0.092_{\pm0.003}$ | $\mathbf{0.034}_{\pm0.002}$ |
| | Depth-Ens (13) | $\mathbf{-0.143}_{\pm0.003}$ | $0.044_{\pm0.000}$ | $0.068_{\pm0.001}$ | $\mathbf{0.006}_{\pm0.002}$ |
| | Depth-Ens (5) | $-0.212_{\pm0.003}$ | $\mathbf{0.064}_{\pm0.001}$ | $\mathbf{0.096}_{\pm0.001}$ | $0.010_{\pm0.001}$ |
| | Dropout | $-0.211_{\pm0.004}$ | $0.051_{\pm0.002}$ | $0.081_{\pm0.002}$ | $0.028_{\pm0.002}$ |
| | Dropout ($p = 0.3$) | $-0.222_{\pm0.006}$ | $0.053_{\pm0.001}$ | $0.084_{\pm0.001}$ | $0.027_{\pm0.002}$ |
| | Ensemble | $-0.145_{\pm0.002}$ | $\mathbf{0.042}_{\pm0.001}$ | $\mathbf{0.063}_{\pm0.001}$ | $0.010_{\pm0.001}$ |
| | SGD | $-0.234_{\pm0.008}$ | $0.051_{\pm0.001}$ | $0.084_{\pm0.002}$ | $0.033_{\pm0.001}$ |
| | S-ResNet | $-0.233_{\pm0.009}$ | $0.057_{\pm0.001}$ | $0.090_{\pm0.002}$ | $0.030_{\pm0.003}$ |
| CIFAR100 | DUN | $\mathbf{-1.182}_{\pm0.018}$ | $0.246_{\pm0.001}$ | $\mathbf{0.377}_{\pm0.001}$ | $\mathbf{0.135}_{\pm0.001}$ |
| | Depth-Ens (13) | $\mathbf{-0.796}_{\pm0.003}$ | $\mathbf{0.210}_{\pm0.001}$ | $0.297_{\pm0.001}$ | $\mathbf{0.015}_{\pm0.002}$ |
| | Depth-Ens (5) | $-1.046_{\pm0.011}$ | $\mathbf{0.264}_{\pm0.002}$ | $0.365_{\pm0.002}$ | $0.038_{\pm0.001}$ |
| | Dropout | $-1.057_{\pm0.020}$ | $0.233_{\pm0.003}$ | $0.347_{\pm0.004}$ | $0.110_{\pm0.002}$ |
| | Dropout ($p = 0.3$) | $-1.088_{\pm0.007}$ | $0.243_{\pm0.002}$ | $0.358_{\pm0.003}$ | $0.111_{\pm0.002}$ |
| | Ensemble | $-0.821_{\pm0.006}$ | $0.204_{\pm0.001}$ | $\mathbf{0.292}_{\pm0.001}$ | $0.047_{\pm0.002}$ |
| | SGD | $-1.155_{\pm0.014}$ | $0.234_{\pm0.002}$ | $0.363_{\pm0.004}$ | $0.133_{\pm0.002}$ |
| | S-ResNet | $-1.108_{\pm0.034}$ | $0.248_{\pm0.004}$ | $0.368_{\pm0.008}$ | $0.113_{\pm0.008}$ |
| Fashion | DUN | $-0.245_{\pm0.029}$ | $0.051_{\pm0.001}$ | $\mathbf{0.087}_{\pm0.002}$ | $0.035_{\pm0.002}$ |
| | Depth-Ens (13) | $\mathbf{-0.141}_{\pm0.002}$ | $\mathbf{0.041}_{\pm0.001}$ | $\mathbf{0.064}_{\pm0.000}$ | $\mathbf{0.011}_{\pm0.001}$ |
| | Depth-Ens (5) | $-0.152_{\pm0.002}$ | $0.044_{\pm0.001}$ | $0.069_{\pm0.001}$ | $0.014_{\pm0.001}$ |
| | Dropout | $-0.208_{\pm0.006}$ | $0.050_{\pm0.001}$ | $0.081_{\pm0.001}$ | $\mathbf{0.050}_{\pm0.000}$ |
| | Dropout ($p = 0.3$) | $-0.185_{\pm0.004}$ | $0.049_{\pm0.001}$ | $0.079_{\pm0.002}$ | $\mathbf{0.050}_{\pm0.000}$ |
| | Ensemble | $-0.180_{\pm0.006}$ | $0.044_{\pm0.001}$ | $0.070_{\pm0.002}$ | $\mathbf{0.050}_{\pm0.000}$ |
| | SGD | $\mathbf{-0.272}_{\pm0.006}$ | $0.051_{\pm0.001}$ | $\mathbf{0.087}_{\pm0.002}$ | $0.048_{\pm0.000}$ |
| | S-ResNet | $-0.217_{\pm0.025}$ | $\mathbf{0.052}_{\pm0.003}$ | $0.084_{\pm0.007}$ | $0.025_{\pm0.006}$ |
| MNIST | DUN | $-0.015_{\pm0.004}$ | $0.004_{\pm0.000}$ | $0.006_{\pm0.000}$ | $0.005_{\pm0.003}$ |
| | Depth-Ens (13) | $\mathbf{-0.009}_{\pm0.000}$ | $\mathbf{0.003}_{\pm0.000}$ | $\mathbf{0.004}_{\pm0.000}$ | $\mathbf{0.001}_{\pm0.000}$ |
| | Depth-Ens (5) | $\mathbf{-0.009}_{\pm0.000}$ | $\mathbf{0.003}_{\pm0.000}$ | $0.005_{\pm0.000}$ | $\mathbf{0.001}_{\pm0.000}$ |
| | Dropout | $-0.011_{\pm0.000}$ | $0.004_{\pm0.000}$ | $0.005_{\pm0.000}$ | $0.040_{\pm0.022}$ |
| | Dropout ($p = 0.3$) | $-0.010_{\pm0.001}$ | $\mathbf{0.003}_{\pm0.000}$ | $0.005_{\pm0.000}$ | $0.030_{\pm0.027}$ |
| | Ensemble | $-0.010_{\pm0.000}$ | $\mathbf{0.003}_{\pm0.000}$ | $0.005_{\pm0.000}$ | $\mathbf{0.050}_{\pm0.000}$ |
| | SGD | $-0.012_{\pm0.001}$ | $0.004_{\pm0.000}$ | $0.006_{\pm0.000}$ | $\mathbf{0.050}_{\pm0.000}$ |
| | S-ResNet | $\mathbf{-0.031}_{\pm0.008}$ | $\mathbf{0.005}_{\pm0.004}$ | $\mathbf{0.011}_{\pm0.005}$ | $0.015_{\pm0.006}$ |
| SVHN | DUN | $\mathbf{-0.202}_{\pm0.021}$ | $\mathbf{0.046}_{\pm0.005}$ | $\mathbf{0.074}_{\pm0.007}$ | $0.019_{\pm0.004}$ |
| | Depth-Ens (13) | $\mathbf{-0.114}_{\pm0.001}$ | $\mathbf{0.027}_{\pm0.000}$ | $\mathbf{0.042}_{\pm0.000}$ | $\mathbf{0.005}_{\pm0.000}$ |
| | Depth-Ens (5) | $-0.129_{\pm0.001}$ | $0.030_{\pm0.000}$ | $0.047_{\pm0.000}$ | $0.006_{\pm0.000}$ |
| | Dropout | $-0.162_{\pm0.014}$ | $0.036_{\pm0.008}$ | $0.057_{\pm0.011}$ | $\mathbf{0.050}_{\pm0.000}$ |
| | Dropout ($p = 0.3$) | $-0.138_{\pm0.002}$ | $0.031_{\pm0.001}$ | $0.049_{\pm0.001}$ | $\mathbf{0.050}_{\pm0.000}$ |
| | Ensemble | $-0.123_{\pm0.002}$ | $\mathbf{0.027}_{\pm0.000}$ | $0.043_{\pm0.000}$ | $\mathbf{0.050}_{\pm0.000}$ |
| | SGD | $-0.177_{\pm0.003}$ | $0.033_{\pm0.001}$ | $0.055_{\pm0.001}$ | $0.049_{\pm0.000}$ |
| | S-ResNet | $-0.168_{\pm0.005}$ | $0.035_{\pm0.001}$ | $0.056_{\pm0.001}$ | $0.018_{\pm0.001}$ |

Figure 23 compares methods' LL performance vs batch time on increasingly corrupted CIFAR10 test data. DUNs are competitive in all cases but their relative performance increases with corruption severity. Dropout shows a clear drop in LL when using a drop rate of 0.3. Figure 24 shows rejection classification plots for CIFAR10 and CIFAR100 vs SVHN

and for Fashion MNIST vs MNIST and KMNIST. Table 11 shows AUC-ROC values for entropy based in-distribution vs OOD classification with all methods under consideration.

Figure 23: Pareto frontiers showing LL for all CIFAR10 corruptions vs batch prediction time. Batch size is 256, split over 2 Nvidia P100 GPUs. Annotations show ensemble elements and Dropout samples. Note that a single element ensemble is equivalent to SGD.

In some cases, similarly to Section 4, we find that using the exact posterior in DUNs is necessary to reduce underconfidence in-distribution. We further investigate this by plotting the posterior probabilities produced by VI, exact inference with batch-norm in train mode and exact inference with batch-norm in test mode in Figure 25. All three approaches assign most probability mass to the final three layers. However, exact inference with BN in test mode differs in that it assigns vanishing low probability mass to all other layers. This is in contrast to VI and train mode BN, where the probability mass assigned to each shallower layer decreases gradually. Predictions are made with BN in test mode. This changes the BN layers from using batch statistics for normalisation to using weighed moving averages computed from the train set. Figure 25 suggests that this shift in network behavior results in predictions from earlier layers being worse and weighing them too heavily in our predictive posterior results in underconfidence.

In Section 3.1, Section 4.1 and Appendix B, we discussed how DUNs trade off expressively and explanation diversity automatically. Figure 26 shows confirms that this mechanism is due to earlier layers obtaining low accuracy. These layers perform representation learning instead. In turn, these layers are assigned low posterior probabilities such that they contribute negligibly to predictions.

Figure 24: Rejection-classification plots. The black line denotes the theoretical maximum performance; all in-distribution samples are correctly classified and OOD samples are rejected first.

Table 11: AUC-ROC values obtained for predictive entropy based separation of in and out of distribution test sets. Bold blue text denotes the best mean value for each dataset and each metric. Bold red text denotes the worst mean value. Note that in some cases the best/worst mean values are within error of other mean values.

| SOURCE TARGET | CIFAR10 | CIFAR100 | FASHION | | MNIST | SVHN |
| | SVHN | | KMNIST | MNIST | FASHION | CIFAR10 |
|---|---|---|---|---|---|---|
| DUN | $0.84_{\pm0.06}$ | $0.76_{\pm0.04}$ | $0.95_{\pm0.01}$ | $0.95_{\pm0.01}$ | $0.86_{\pm0.03}$ | $0.92_{\pm0.02}$ |
| DUN (exact) | $0.90_{\pm0.03}$ | $0.77_{\pm0.03}$ | $0.94_{\pm0.01}$ | $0.95_{\pm0.01}$ | $0.87_{\pm0.03}$ | $0.90_{\pm0.03}$ |
| Depth-Ens (13) | $0.91_{\pm0.00}$ | $0.80_{\pm0.01}$ | $0.97_{\pm0.00}$ | $0.96_{\pm0.00}$ | $0.99_{\pm0.00}$ | $0.98_{\pm0.00}$ |
| Depth-Ens (5) | $0.84_{\pm0.02}$ | $0.79_{\pm0.02}$ | $0.97_{\pm0.00}$ | $0.96_{\pm0.00}$ | $0.94_{\pm0.04}$ | $0.97_{\pm0.00}$ |
| Dropout | $0.90_{\pm0.01}$ | $0.75_{\pm0.05}$ | $0.96_{\pm0.01}$ | $0.97_{\pm0.01}$ | $0.95_{\pm0.04}$ | $0.94_{\pm0.01}$ |
| Dropout (0.3) | $0.88_{\pm0.01}$ | $0.76_{\pm0.03}$ | $0.96_{\pm0.01}$ | $0.96_{\pm0.01}$ | $0.91_{\pm0.06}$ | $0.95_{\pm0.01}$ |
| Ensemble | $0.93_{\pm0.02}$ | $0.77_{\pm0.01}$ | $0.96_{\pm0.00}$ | $0.96_{\pm0.00}$ | $0.98_{\pm0.00}$ | $0.97_{\pm0.00}$ |
| S-ResNet | $0.87_{\pm0.05}$ | $0.79_{\pm0.03}$ | $0.96_{\pm0.01}$ | $0.96_{\pm0.01}$ | $0.86_{\pm0.04}$ | $0.93_{\pm0.01}$ |
| SGD | $0.89_{\pm0.02}$ | $0.76_{\pm0.03}$ | $0.95_{\pm0.01}$ | $0.96_{\pm0.01}$ | $0.94_{\pm0.04}$ | $0.93_{\pm0.01}$ |

Figure 25: Posterior probabilities produced by VI, exact inference with batch-norm in train mode and exact inference with batch-norm in test mode on the CIFAR10 train-set.

Figure 26: Comparison of per-depth error rates for the layers of a DUN and depth-ensemble elements of the same depth.

# G   DUNs for Neural Architecture Search

In this section, we briefly explore the application of DUNs to Neural Architecture Search (NAS) and how architecture hyperparameters affect the posterior over depth. This section is based on the previous work (Antorán et al., 2020). Please see that paper for more information, including further experimental evaluation and analysis as well as contextualisation of this technique in the NAS literature.

After training a DUN, as described in Section 3.2, $q_{\boldsymbol{\alpha}}(d{=}i) = \alpha_i$ represents our confidence that the number of blocks we should use is $i$. We would like to use this information to prune our network such that we reduce computational cost while maintaining performance. Recall our training objective (2):

$$\mathcal{L}(\boldsymbol{\alpha}, \boldsymbol{\theta}) = \sum_{n=1}^{N} \mathbb{E}_{q_{\boldsymbol{\alpha}}(d)} \left[ \log p(\mathbf{y}^{(n)} | \mathbf{x}^{(n)}, d; \boldsymbol{\theta}) \right] - \mathrm{KL}(q_{\boldsymbol{\alpha}}(d) \, \| \, p_{\boldsymbol{\beta}}(d)).$$

In low data regimes, where both the log-likelihood and KL divergence terms are of comparable scale, we obtain a posterior with a clear maximum. We choose

$$d_{\mathrm{opt}} = \arg\max_i \alpha_i. \tag{11}$$

as our fixed depth. In medium-to-big data regimes, where the log-likelihood dominates our objective, we find that the values of $\alpha_i$ flatten out after reaching an appropriate depth. For examples of this phenomenon, compare the approximate posteriors over depth shown in Figure 27 and Figure 32. We propose a heuristics for choosing $d_{\mathrm{opt}}$ in this case. We choose the smallest depth with a probability larger that 95% of the maximum of $q$:

$$d_{\mathrm{opt}} = \min_i \{ i : q(d{=}i) \geq 0.95 \max_j q(d{=}j) \}. \tag{12}$$

Both heuristics aim to keep the minimum number of blocks needed to explain the data well. We prune all blocks after $d_{opt}$ by setting $q_{\boldsymbol{\alpha}}(d{=}d_{\mathrm{opt}}) = q_{\boldsymbol{\alpha}}(d{\geq}d_{\mathrm{opt}})$ and then $q_{\boldsymbol{\alpha}}(d{>}d_{\mathrm{opt}}) = 0$. Instead of also discarding the learnt probabilities over shallower networks, we incorporate them when making predictions on new data points $\mathbf{x}^*$ through marginalisation:

$$p(\mathbf{y}^*|\mathbf{x}^*) \approx \sum_{i=0}^{d_{\mathrm{opt}}} p(\mathbf{y}^*|\mathbf{x}^*, d{=}i; \boldsymbol{\theta}) q_{\boldsymbol{\alpha}}(d{=}i). \tag{13}$$

We refer to pruned DUNs as Learnt Depth Networks (LDNs) and study them, contrasting them with (standard) Determinisitc Depth Networks (DDNs) in the following experiments.

## G.1   Toy Experiments

We generate a 2d training set by drawing 200 samples from a 720° rotation 2-armed spiral function with additive Gaussian noise of $\sigma = 0.15$. The test set is composed of an additional 1800 samples. Choosing a relatively small width for each hidden layer $w = 20$ to ensure the task can not be solved with a shallow model, we train fully-connected LDNs with varying maximum depths $D$ and DDNs of all depths up to $D{=}100$. Figure 27 shows how the depths to which LDNs assign larger probabilities match those at which DDNs perform best. Predictions from LDNs pruned to $d_{opt}$ layers outperform DDNs at all depths. The chosen $d_{opt}$ remains stable for increasing maximum depths up to $D \approx 50$. The same is true for test performance, showing some robustness to overfitting. After this point, training starts to become unstable. We repeat experiments 6 times and report standard deviations as error bars.

We further explore the properties of LDNs in the small data regime by varying the layer width $w$. As shown in Figure 28, very small widths result in very deep LDNs and worse test performance. Increasing layer width gives our models more representation capacity at each layer, causing the optimal depth to decrease rapidly. Test performance remains stable for widths in the range of 20 to 500, showing that our approach adapts well to changes in this parameter. The test log-likelihood starts to decrease for widths of 1000, possibly due to training instabilities.

Setting $w$ back to 20, we generate spiral datasets with varying degrees of rotation while keeping the number of train points fixed to 200. In Figure 31, we see how LDNs increase

Figure 27: Left: posterior over depths for a LDN of $D = 50$ trained on our spirals dataset. Test log-likelihood values obtained for DDNs at every depth are overlaid with the log-likelihood value obtained with a LDN when marginalising over $d_{opt} = 9$ layers. Right: the LDN's depth, chosen using (11), and test performance remain stable as $D$ increases up until $D \approx 50$.

Figure 28: Evolution of LDNs' chosen depth and test performance as their layer width $w$ increases. The results obtained when making predictions by marginalising over all $D = 20$ layers overlap with those obtained when only using the first $d_{opt}$ layers. The x-axis is presented in logarithmic scale.

their depth to match the increasing complexity of the underlying generative process of the data. For rotations larger than 720°, $w = 20$ may be excessively restrictive. Test performance starts to suffer significantly. Figure 29 shows how our LDNs struggle to fit these datasets well.

Returning to 720°spirals, we vary the number of training points in our dataset. We plot the LDNs' learnt functions in Figure 30. LDNs overfit the 50 point train set but, as displayed in Figure 29, learn very shallow network configurations. Increasingly large training sets allow the LDNs to become deeper while increasing test performance. Around 500 train points seem to be enough for our models to fully capture the generative process of the data. After this point $d_{opt}$ oscillates around 11 layers and the test performance remains

Figure 29: Functions learnt at each depth of a LDN on increasingly complex spirals. Note that single depth settings are being evaluated in this plot. We are not marginalising all layers up to $d_{opt}$.

Figure 30: Functions learnt by LDNs trained on increasingly large spiral datasets. Note that single depth settings are being evaluated in this plot. We are not marginalising all layers up to $d_{opt}$.

Figure 31: The left-side plots show the evolution of test performance and learnt depth as the data complexity increases. The right side plots show changes in the same variables as the number of train points increases. The results obtained when making predictions by marginalising over all $D = 20$ layers overlap with those obtained when only using the first $d_{opt}$ layers. Best viewed in colour.

constant. Marginalising over $D$ layers consistently produces the same test performance as only considering the first $d_{opt}$. All figures are best viewed in colour.

## G.2 Small Image Datasets

We further evaluate LDNs on MNIST, Fashion-MNIST and SVHN. Note that the network architecture used for these experiments is different from that used for experiments on the same datasets in Section 4.4 and Appendix F.3. It is described below. Each experiment is repeated 4 times to produce error bars. The results obtained with $D = 50$ are shown in Figure 32. The larger size of these datasets diminishes the effect of the prior on the ELBO. Models that explain the data well obtain large probabilities, regardless of their depth. For MNIST, the probabilities assigned to each depth in our LDN grow quickly and flatten out around $d_{opt} \approx 18$. For Fashion-MNIST, depth probabilities grow slower. We obtain $d_{opt} \approx 28$. For SVHN, probabilities flatten out around $d_{opt} \approx 30$. These distributions and $d_{opt}$ values correlate with dataset complexity. In most cases, LDNs achieve test log-likelihoods competitive with the best performing DDNs.

Figure 33 shows more detailed experiments comparing LDNs with DDNs on image datasets. We introduce expected depth $d_{opt} = \text{round}(\mathbb{E}_{q_\alpha}[d])$ as an alternative to the 95th percent heuristic introduced above. The first row of the figure adds further evidence that the depth learnt by LDNs corresponds to dataset complexity. For any maximum depth, and both pruning approaches, the LDN's learnt depth is smaller for MNIST than Fashion-MNIST and likewise smaller for Fashion-MNIST than SVHN. For MNIST, Fashion-MNIST and, to a lesser extent, SVHN the depth given by the 95th percent pruning tends to saturate. On the other hand, the expected depth grows with $D$, making it a less suitable pruning strategy.

Figure 32: Approximate posterior over depths for LDNs of $D=50$ trained on image datasets. Test log-likelihoods obtained for DDNs at various depths are overlaid with those from our LDNs when marginalising over the first $d_{opt}$ layers. The depth was chosen using (12)

As shown in rows 2 to 5, for SVHN and Fashion-MNIST, 95th percentile-pruned LDNs suffer no loss in predictive performance compared to expected depth-pruned and even non-pruned LDNs. They outperform DDNs. For MNIST, 95th percent pruning gives results with high variance and reduced predictive performance in some cases. Here, DDNs yield better log-likelihood and accuracy results. Expected depth is more resilient in this case, matching full-depth LDNs and DDNs in terms of accuracy.

## G.3 Experimental Setup for NAS

For experiments on the spirals dataset, our input $f_0$ and output $f_{D+1}$ blocks consist of linear layers. These map from input space to the selected width $w$ and from $w$ to the output size respectively. Thus, selecting $d=0 \Rightarrow b_i=0 \, \forall i \in [1, D]$ results in a linear model. The functions applied in residual blocks, $f_i(\cdot) \, \forall i \in [1, D]$, consist of a fully connected layer followed by a ReLU activation function and Batch Normalization (Ioffe and Szegedy, 2015).

Our architecture for the image experiments uses a 5×5 convolutional layer together with a 2×2 average pooling layer as an input block $f_0$. No additional down-sampling layers are used. The output block, $f_{D+1}$, is composed of a global average pooling layer followed by a fully connected residual block, as described in the previous paragraph, and a linear layer. The function applied in the residual blocks, $f_i(\cdot) \, \forall i \in [1, D]$, matches the preactivation bottleneck residual function described by He et al. (2016b) and uses 3×3 convolutions. The outer number of channels is set to 64 and the bottleneck number is 32.

Figure 33: Comparisons of DDNs and LDNs using different pruning strategies and maximum depths. *LDN-95* and *LDN-*$\mathbb{E}$ refer to the pruning strategy described in (12) and $d_{\mathrm{opt}} = \mathrm{round}(\mathbb{E}[q_{\boldsymbol{\alpha}}(d)])$, respectively. 1$^{\mathrm{st}}$ row: comparison of $d_{\mathrm{opt}}$. 2$^{\mathrm{nd}}$ row: comparison of test log-likelihoods for DDNs and LDNs with 95 percent pruning. 3$^{\mathrm{rd}}$ row: comparison of test log-likelihoods for LDN pruning methods. 4$^{\mathrm{th}}$ and 5$^{\mathrm{th}}$ rows: as above but for test error.

## H Implementing a DUN

In this section we demonstrate how to implement a DUNs computational model by modifying standard feed-forward NNs written in PyTorch. First, we show this for a simple MLP and then for the more realistic case of a ResNet, starting from the default PyTorch implementation.

### H.1 Multi-Layer Perceptron

Converting a simple MLP to a DUN requires only around 8 lines of changes, depending on the specific implementation. Only 4 of these changes, in the `forward` function, are significant differences. The following listing shows the `git diff` of a MLP implementation before and after being converted.

```
 import torch
 import torch.nn as nn

 class MLP(nn.Module):
     def __init__(self, input_dim, hidden_dim, output_dim, num_layers):
         super(MLP, self).__init__()

+        self.output_dim = output_dim

         layers = [nn.Sequential(nn.Linear(input_dim , hidden_dim),
                     nn.ReLU())]

         for _ in range(num_layers):
             layers.append(nn.Sequential(nn.Linear(hidden_dim, hidden_dim),
                             nn.ReLU()))

-        layers.append(nn.Linear(hidden_dim, output_dim))
+        self.output_layer = nn.Linear(hidden_dim, output_dim)

-        self.layers = nn.Sequential(*layers)
+        self.layers = nn.ModuleList(layers)

     def forward(self, x):
+        act_vec = x.new_zeros(len(self.layers), x.shape[0], self.output_dim)

+        for idx, layer in enumerate(self.layers):
+            x = self.layers[idx](x)
+            act_vec[idx] = self.output_layer(x)

-        return self.layers(x)
+        return act_vec
```

### H.2 PyTorch ResNet

To convert the official PyTorch ResNet implementation[5] into a DUN, we just need to make 17 changes. Many of these changes involve changing only a few characters on each line. Rather than looking at the whole file, which is over 350 lines long, we'll look only at the changes.

The first change that needs to be made is to the `_make_layer` function on line 177 of `resnet.py`. This function now needs to return a list of layers rather than a `nn.Sequential` container.

```
                 base_width=self.base_width, dilation=self.dilati
                 norm_layer=norm_layer))

- return nn.Sequential(*layers)
+ return layers
```

With that change, we can modify the `__init__` function of the `ResNet` class on line 124 of `resnet.py`. We will create a `ModuleList` container to hold all of the layers of the ResNet. This change has been made so that our `forward` function has access to the each layer individually.

```
                              dilate=replace_stride_with_dilation[1]
  self.layer4 = self._make_layer(block, 512, layers[3], stride=2,
                              dilate=replace_stride_with_dilation[2]
+ self.layers = nn.ModuleList(self.layer1 + self.layer2 +
+                             self.layer3 + self.layer4)
```

Before implementing the forward function, we need to implement the adaption layers that ensure that inputs to the output block always have the correct number of filters. This is also done in the `__init__` function. Each adaption layer up-scales the number of filters by a factor of 2. Some layers need to have their outputs up-scaled multiple times which is kept track of by `self.num_adaptions`.

```
 self.avgpool = nn.AdaptiveAvgPool2d((1, 1))
 self.fc = nn.Linear(512 * block.expansion, num_classes)

+ self.num_adaptions = [0] * layers[0] + [1] * layers[1] + \
+                      [2] * layers[2] + [3] * layers[3]
+ adapt0 = nn.Sequential(
+     conv1x1(64*block.expansion, 128*block.expansion, stride=2),
+     self._norm_layer(128*block.expansion), self.relu)
+ adapt1 = nn.Sequential(
+     conv1x1(128*block.expansion, 256*block.expansion, stride=2),
+     self._norm_layer(256*block.expansion), self.relu)
+ adapt2 = nn.Sequential(
+     conv1x1(256*block.expansion, 512*block.expansion, stride=2),
+     self._norm_layer(512*block.expansion), self.relu)
+ adapt3 = nn.Identity()
+ self.adapt_layers = nn.Sequential(adapt0, adapt1, adapt2, adapt3)
```

The changes to the `_forward_impl` function on line 201 of `resnet.py` involve iterating over the layer list, up scaling layer outputs, and saving all of the activations of the output block.

```
 x = self.relu(x)
 x = self.maxpool(x)

- x = self.layer1(x)
- x = self.layer2(x)
- x = self.layer3(x)
- x = self.layer4(x)
+ act_vec = x.new_zeros(len(self.layers), x.shape[0], self.n_classes)
+ for layer_idx, layer in enumerate(self.layers):
+     x = layer(x)
+     y = self.adapt_layers[self.num_adaptions[layer_idx]:](x)
- x = self.avgpool(x)
+     y = self.avgpool(y)
- x = torch.flatten(x, 1)
+     y = torch.flatten(y, 1)
- x = self.fc(x)
+     y = self.fc(y)
+     act_vec[layer_idx] = y

- return x
+ return act_vec
```

The final change is to store the number of classes in the `__init__` function so that the `_forward_impl` function can pre-allocate a tensor of the correct size.

```
 self.inplanes = 64
 self.dilation = 1
```

```
+ self.n_classes = num_classes
```

# I  Negative Results

Here we briefly discus some ideas that seemed promising but were ultimately dead-ends.

**Non-local Priors**  These priors are ones which have zero density in the region of the *null value* (often zero). Examples of such priors include the pMOM, piMOM, and peMOM priors (Johnson and Rossell, 2012; Rossell et al., 2013), shown in Figure 34.

We attempted to train DUNs with these priors, hoping that enforcing that each weight in the network was non-zero would, in turn, force each block of the DUN to make a significantly different prediction to the previous block. Unfortunately training with non-local priors was unstable and resulted in poor performance.

Figure 34: Comparison of non-local priors with the standard normal distribution.

**MLE training**  As described in Appendix B, MLL training on DUNs tends to get stuck in local optima in which the posterior over depth collapses to a single arbitrary depth. In practice we found that VI training greatly reduces this problem.

**Concat Pooling**  This technique combines the average and max pooling operations by concatenating their results. We tried to apply it before the final linear layer in ResNet-50. We suspected that for DUNs based on ResNets this would be useful because the output block needs to work for predictions at multiple resolutions. Unfortunately, we found that the extra information provided by concat pooling over the standard average pooling resulted in strong overfitting.

**Scaling ResNet-50 to ImageNet**  We trained a ResNet-50 DUN on the ImageNet dataset. However, in line with Havasi et al. (2020), we found that a ResNet-50 does not have enough capacity to provide multiple explanations of the complex ImageNet dataset. As a result, the depth posterior for ResNet-50 invariably collapsed to a delta distribution. DUN performance on ImageNet is poorer than standard SGD training, as shown in Figure 35.

Figure 35: Error and LL for ImageNet at varying degrees of corruption. Due to computational costs only a single model was trained and evaluated for each method. As a result, we do not provide error bars.

Note that, while it is clear that DUN performance in this setting is not strong, we only had enough computational resources to train each model one time. Without error bars, it is difficult to draw strong conclusions about the results. We hypothesize that a more heavily over-parameterised ResNet variant, such as ResNet-152 or a wide ResNet-50, would be able to support a DUN.

## Footnotes

[3]For ResNet-50, which contains 23.52 M parameters, a DUN over the first 13 blocks (using $1 \times 1$ convolution adaption layers) contains 26.28 M parameters which is an increase of only 1.17%.

[4]A low rank dense layer with input of size $n_1$ and output of size $n_2$ can be constructed by composing two standard dense layers of size $n_1 \times n_3$ and $n_3 \times n_2$ where $n_3 << n_1, n_2$.

[5] https://github.com/pytorch/vision/blob/master/torchvision/models/resnet.py