[Reviews · NeurIPS 2020]

Review 1

Summary and Contributions: In this paper, the authors proposed to impose Bayesian inference on the depth of deep networks to accommodate the pain of choosing network depth. The proposed Deep Uncertainty Network (DUN) is conceptually easy and computationally cheap to implement. The authors also verified that it improves calibration and out-of-distribution performance on UCI regression and image classification.

Strengths: If we have the budget to train an ensemble of deep networks with different depths, how to choose the individual depth to achieve the maximum benefits is an important question. This paper accommodates this question by a theoretically sound approach, which also enjoys cheap computation and implementation easiness. Traditional Bayesian approach to improve uncertainty performance requires multiple samples, leading to the painful need of multiple forward passes. This bottlenecks the use of many uncertainty methods (such as MC-dropout) in practice because the inference cost is linearly increasing as the number of samples needed. One obvious advantage of DUN is that multiple samples (depth uncertainty) can be computed in a deep network within just one forward pass. This gives its great potential to be successfully employed in real-world application. Additionally, the dimension of variable (depth) in Bayesian inference is just 1. This also removes the need for multiple Monte Carlo samples. This paper provides a very detailed implementation in the appendix. For example, it compares two different objectives (VI and MLL) and illustrates that MLL leads to collapsed single depth. It also listed a number of promising design choices which did not work.

Weaknesses: My main concern is mainly about the experiments section. It is great to see the demonstration of the method on ResNet-50. But I couldn’t find the comparison of test accuracy through the paper. There is only log-likelihood included in the appendix. I would expect a comparison on test accuracy among DUN, MC-dropout, deep ensembles, deep ensembles with different depths. It would make the experiments stronger if the authors include a comparison to stochastic depth network [1]. Sampling stochastic depth multiple times also represents uncertainty over depth. Even though this leads to computational overhead compared to DUN, it is still interesting to see how they compare in terms of test accuracy and OOD performance. How does D determine for each type of architecture and dataset? This might be a critical choice because it plays an important role in the regularization term, which prevents DUN to choose the deepest depth? Because we know the deeper and wider networks always perform better? It is also interesting if we can visualize the posterior distribution over the depths. Another concern I have is for a very deep network, its intermediate layers can learn some low-level representations at its free will. But in DUQ, many intermediate layers are directly connected to the classification module. This limits the expressive power of intermediate layers and might compromise the prediction accuracy of the deepest sub-network in DUN. It would be better if authors can make a study on the individual test accuracy of each sub-network. Diversity analysis among these sub-networks is also good for explaining the gain of DUN on OOD performance. Since every sub-network is connected to a classification layer, this implicitly adds some parameters. Fully connected layers are the most parameter-dense layer in ResNet, so a comparison of DUN and other methods on parameter count is also important. [1]: Gao Huang, Yu Sun, Zhuang Liu, Daniel Sedra, and Kilian Q Weinberger. Deep networks with stochastic depth. ECCV 2016

Correctness: The claims are rigoriously correct. I also roughly checked the code and it seems consistent to the methods described in the paper. Minor comments: There are some regions in figure 1 having infinite uncertainty estimation. An explanation on this complements the figure. The claim that all methods except DUN require multiple forward passes is not strictly correct. [2] and [3] are batch friendly methods which also show improvements on uncertainty estimates. [2]: Dusenberry, Michael W., Ghassen Jerfel, Yeming Wen, Yi-An Ma, Jasper Snoek, Katherine A. Heller, Balaji Lakshminarayanan and Dustin Tran. “Efficient and Scalable Bayesian Neural Nets with Rank-1 Factors.” ICML 2020 [3]: Wen, Yeming, Dustin Tran and Jimmy Ba. “BatchEnsemble: An Alternative Approach to Efficient Ensemble and Lifelong Learning.” ICLR 2020

Clarity: Yes.

Relation to Prior Work: This paper discusses a fair amount of related works.

Reproducibility: Yes

Additional Feedback: Updated reviews: after reading the rebuttal, most of my concerns are addressed. Some of them such as the accuracy/calibration on CIFAR-10/100 and their corruptions performance still remain. I raise my score to 6.


Review 2

Summary and Contributions: The paper looks at the problem of uncertainty estimation in deep neural networks. The authors propose to treat the depth of a deep network as a random variable and marginalise over predictions made on features of each level to make a prediction. This procedure requires minimal changes to training a standard network and incurs little overhead at test time. There is extensive experimental evaluation of the proposed method, showing that it performs well in a range of situations.

Strengths: The paper is very well written with clear explanations and visualisations of the method (e.g. figure 3 is very nice). The proposed algorithm is simple, and seemingly effective at solving an important problem. I found it interesting to see EM/MLL perform so poorly compared to VI and I enjoyed the additional analysis in the supplementary materials.

Weaknesses: My main concerns is the significance of the contribution. Given that deep ensembles, mc dropout, and stochastic batch norm are known to increase uncertainty quality, it is not suprising that marginalising over depth also works. However it has not been done before, as far as I know, which means it is novel. The authors claim *calibrated* uncertainty (line 41), meaning that high uncertainty is correlated with low accuracy. In figure 6, the OOD rejection plot, the authors show that DUN does not smoothly go to 1.0 accuracy but instead is weirdly flat after 75% rejected (i.e. all of SVHN removed) this indicates that DUN's uncertainty on the remaining CIFAR10 images is not well calibrated. Otherwise it would reach 1.0 accuracy much earlier. The authors argue that DUN is "underconfident" but I don't understand how that would explain this graph. It is fine if uncertain data points from CIFAR10 are rejected before SVHN, as long as they're wrongly classified (i.e. the model is calibrated). If you were to do OOD evaluated only (e.g. using AUROC) then I would agree with the argument, but this is a rejection classification plot. I am also curious how the depth weighting changes between the learned values (Variational parameters) and the exact parameters which require the additional run over the training set. I would be curious to see more evidence of the calibration claim and a discussion of the flat DUN result from 75% onwards (and figure 24, supplementary materials, CIFAR 100 result). In lines 252-257: the authors discuss compute time and claim that their method is 1.02 slower than the standard network, but somehow equivalent to only 0.47 size ensemble on average because of loading the ensembles in memory. This seems misleading because that depends on a particular engineering decision and is not a general fact. It seems more fair to set a single dropout sample, and a single ensemble model at 1, and increase these linearly with ensemble/sample size (DUN would still do well under this setting). Also this would remove some of the inconsistent gaps in the right bottom graph of figure 6. It is very unclear why ensemble 2 takes long, but ensemble 3 is fast again. Nit: the authors claim exact predictive posteriors in a single forward pass (line 40), generally this is implied to mean posterior over model weights however the authors mean posterior over depth variable. It would be better to be explicit about this fact.

Correctness: Yes

Clarity: Very well, a joy to read.

Relation to Prior Work: Yes, related work is discussed and compared to.

Reproducibility: Yes

Additional Feedback: I think this is a simple idea very well executed. The authors amply compensate the limited novelty with rigorous experimentation and clear explanations. After author response: My main concern was with the calibration of the uncertainty. I am still not fully convinced that figure d) gives an answer to this and I recommend the authors to verify or tone down this claim. Other than that the rebuttal was well written and informative. I've increased my score to an 8.


Review 3

Summary and Contributions: This paper proposes a method for uncertainty estimation with deep neural networks. The authors propose a probabilistic model called DUNs (Depth Uncertainty Networks), where the depth of the network is a latent variable. Inference in the proposed model can be efficiently performed with variational inference. The key characteristic of the proposed model is its simplicity: predictive distribution can be computed in a single forward pass; the weights of the network are treated as parameters and the inference performed over a categorical distribution. In the experiments, the predictive accuracy and the quality of uncertainty estimation with DUNs are evaluated on tabular regression datasets and image classification datasets. DUNs are compared with MC-dropout, deep ensembles and Mean-Field Variational Inference. === Update === I thank the authors for addressing my and other reviewers' questions in the authors' response. The authors have addressed my concern about expressive power of the proposed architecture. The authors point out that the model can use lower layers for representation learning and the assign lower posterior probabilities to these layers so that they do not hurt the predictive accuracy. I am also satisfied with authors' comments and additional experimental results provided in response to questions raise by other reviewers. After reading other reviews and authors' response, I still think that this is a good paper. I vote for acceptance and keep my original score unchanged. =============

Strengths: The paper is technically sound. The formulated model is clear and well-described. The authors analyse two methods for training of DUNs: direct MLE and Variational Inference (VI) and find the MLE training is prone to local optimas which put all posterior probability mass on a single depth, while training dynamics of VI avoid this problem and also does not suffer from the approximation errors as inference is performed over categorical distribution of depths and the ELBO is tight. The experimental evaluation is comprehensive and the experimental set-ups and evaluation protocols are chosen adequately. All necessary details of the experiments are provided. The proposed method is simple. It can be easily implemented in deep learning frameworks and its computational costs are very low both at training and prediction stages.

Weaknesses: I am a bit concerned about the expressive power of the proposed model. The assumption that predictions at any depth can be computed with the same output block seems limiting. It is not exactly clear how the hidden representations at different levels will be structured given that at all levels the representations have to be adjusted to the same decision boundary defined by the output block. It is not clear to what extent this design limits the expressive power of the model. I would encourage the authors to include a discussion of this issue and provide more experimental results. For instance, it would be good to see the comparison of final test set prediction accuracy and test set NLL on CIFAR-10/CIFAR-100 for standard ResNet-50 and the modified ResNet-50 DUNs architecture.

Correctness: This is a technically strong work. The claims, the methods, and the experimental protocols are correct.

Clarity: The paper is very well written. The model, the training methods, and the experiments are clearly described.

Relation to Prior Work: The paper includes a complete overview of the related work on uncertainty estimation in deep learning, Bayesian neural networks, and ensemble methods. To the best of my knowledge, this is the first paper to use a model that treats the depth of a network as a random variable with application to uncertainty estimation.

Reproducibility: Yes

Additional Feedback:


Review 4

Summary and Contributions: This paper proposes a new method for estimating uncertainty in neural networks. In particular, a) It proposes performing probabilistic inference over the depth of the network, b) It evaluates uncertainties in a single-forward pass, c) It provides uncertainty calibration, out-of-distribution detection, and predictive performance competitive with more computationally expensive baselines.

Strengths: + Perform Bayesian inference in NNs over a single forward pass. + Novelty of treating depth as a random variable for uncertainty measurements and doing exact Bayesian inference. + Competitive performance with prior more-expensive methods requiring multiple passes over the NN

Weaknesses: - Outperformed by methods not restricted to low resources settings like ensembles in a variety of tasks. - Dropout based uncertainity estimation require similar compute time as proposed method, and there is no significant advantage of using this method over dropout. In Table 1, the inference time for dropout is indeed less than DUN. - Uncertainity estimation over network depth might be more limited than estimating uncertainities over the parameter space.

Correctness: Yes, the claims and methods are correct and emperically sound.

Clarity: Yes

Relation to Prior Work: Yes

Reproducibility: Yes

Additional Feedback: I am not fully convinced of the advantage of this method over dropout. Pls address the points listed in the Weaknesses section.

[Author Response · NeurIPS 2020]

**Overall:** We thank the reviewers for their time and insightful comments/suggestions. We are happy that the reviewers
appreciated the novelty and relevance of our contribution: Probabilistic reasoning is performed over NN depth, as
opposed to more common weight space approaches (R1, R3, R4). Competitive uncertainty estimates are obtained
with a single NN forward pass (R1, R2, R3, R4). We are also pleased that the reviewers highlighted that our paper is
clear and detailed (R1, R2, R3), that our method is simple conceptually and in implementation (R1, R2, R3), and the
comprehensiveness of our experimental evaluation (R2, R3). We are glad three reviewers recommend acceptance (R2,
R3, R4). R1 is mainly concerned with our experimental evaluation. We provide new experiments and address others'
concerns below. We will incorporate suggestions and expanded results using the additional page available.

**R1** *Accuracy*: See Fig. 6, top row. For clarity, we will include an appendix table for all datasets. *Baselines*: Results
for ensembles of different depths and stochastic depth resnets (ResNet50, uncertain layers 1-13 in both cases) are in
**a)**. The former requires training multiple NNs and performs similarly to deep ensembles. Both require evaluating
multiple NNs. The latter is a particularisation of MC Dropout, performs worse, and might inherit its limitations (Foong
et. al., 2019). We will update all experiments in Fig. 6 with the new baselines. *Architecture, $D$*: We perform additional
experiments exploring the effect of $D$ and width on the depth posterior. These will be added to the Appendix. To
summarise: larger $D$ provides more opportunity for explanation diversity, and thus increased performance. Past a large
enough $D$, increases yield diminishing returns. For wider blocks or simpler datasets (e.g. MNIST is simpler CIFAR10),
$D$ can be smaller without performance loss. The regularisation impact of $D$ is usually negligible since, unlike the
likelihood, the KL term in the ELBO does not scale with the data (see §B). *Depth posteriors*: Please see Fig. 3, and
Figs. 7-10. *Expressive power*: DUNs trade off expressivity and explanation diversity automatically; Earlier layers
obtain low accuracy's (see **b)**) and are assigned low posterior probabilities (see §3.1). They contribute negligibly to
predictions, performing representation learning instead. In the limit of being capacity constrained, we have observed
the posterior collapse to a delta, recovering a vanilla NN. In practise, NNs have excess modelling capacity. **a)** shows
DUNs performing competitively with baselines given a fixed architecture. *Diversity Analysis:* **c)** shows the mean KL
divergence between different depths' predictions, for DUNs, and different samples', for baselines. DUNs present large
diversity in-distribution, potentially resulting in some underconfidence (Fig 6. bottom left). DUNs' diversity grows
OOD, allowing for robustness. *No. parameters*: DUNs only add parameters where adaption layers are necessary. We
adapt channels in CNNs with 1x1 convs (see §E3.1), which add few parameters. ResNet50: 23.52M weights. ResNet50
DUN (1-13): 26.28M. Increase of 1.17%. Our FC DUNs use constant width. Otherwise, width adaption could be
efficiently implemented with low rank weight matrices. *Fig. 3 large variance*: As NNs are flexible, often underspecified
models, their predictions can diverge OOD. This behaviour is also seen with ensembles (Figs. 4, 13, 19). Will include
discussion. *Batch friendly methods*: Some baselines are parallelisable: i.e. multiple forward passes can be performed
for an input by replicating it across a batch. Our method only requires a single forward pass. We will clarify.

**R2** *Significance*: DUNs are conceptually simple but differ from most previous work in that they are a non weight space
approach. This allows DUNs to sidestep the intractabilities and computational cost often associated with BNNs. DUNs
are orthogonal to, and could be combined with, weight uncertainty. *Rejection-classification plot*: We agree with your
assessment: underconfidence on correctly classified points leads to these being rejected together with OOD / wrongly
classified points, flattening the curve. Requested posteriors are shown in **d)**. Batch norm (BN) seems to be the culprit.
The exact posterior, computed with train mode BN, matches the variational posterior. The test mode BN posterior is
more peaked and fixes underconfidence (Fig. 6). *ResNet50 timing*: We include loading times in our results as storing
multiple ResNet50s in memory is often impractical. Without loading, ensemble times match dropout. We will clarify
this and mention that inconsistent plot gaps are due to "single element" ensembles not considering loading times.

**R3** *Expressive power*: Indeed, a shared output block could be a bottleneck. In practise, we do not observe this to be an
issue. More flexible output blocks actually resulted in overfitting (see §I, Concat Pooling). The depth posterior allows
blocks to specialise on either representation learning or predictions. Please see R1: Accuracy, Expressive power.

**R4** *Comparison to Dropout*: For a fixed architecture, DUNs are always faster than Dropout (see Fig. 6 bottom right).
Our regression experiments use Bayesian optimisation (BO) to choose architectures. In Table 1, the DUN is using a
significantly deeper model. Even with BO, DUNs are most often faster than Dropout (Fig. 5 timing row). We find DUNs
to outperform Dropout in terms of uncertainty estimation in most tasks (Fig. 5 TCE row and Fig. 6). *Limitations*: In
practise, the complexity of weight space posteriors limits these methods' expressivity (Foong et. al., 2019). This can be
seen in §4.2, §F.1. Depth uncertainty is orthogonal to weight uncertainty, side-stepping this issue. Both can be combined.

[Meta-Review · NeurIPS 2020]

This paper proposes to treat depth of the network as a random variable and marginalize over that to achieve better uncertainty. The authors show that this can be performed efficiently in a single forward pass, and demonstrate improved uncertainty estimates on both regression and classification benchmarks (including corrupted versions and out-of-distribution evaluation). The reviewers initially raised several questions particularly on experimental setup, flexibility of the depth posterior and comparisons to stochastic depth and dropout). During the discussion, the reviewers agreed that the author rebuttal satisfactorily addresses the major concerns and some of them raised their scores correspondingly as well. Overall, this is a good paper and I recommend accept.